# REFINE NOW, QUERY FAST: A DECOUPLED REFINEMENT PARADIGM FOR IMPLICIT NEURAL FIELDS

**Tianyu Xiong**[*]
Department of Computer Science and Engineering
Ohio State University
Columbus, OH 43210, USA
`xiong.336@osu.edu`

**Skylar W. Wurster**
Adobe
San Francisco, CA, USA
`swwurster@gmail.com`

**Han-Wei Shen**
Department of Computer Science and Engineering
Ohio State University
Columbus, OH 43210, USA
`shen.94@osu.edu`

## ABSTRACT

Implicit Neural Representations (INRs) have emerged as promising surrogates for large 3D scientific simulations due to their ability to continuously model spatial and conditional fields, yet they face a critical fidelity-speed dilemma: deep MLPs suffer from high inference cost, while efficient embedding-based models lack sufficient expressiveness. To resolve this, we propose the Decoupled Representation Refinement (DRR) architectural paradigm. DRR leverages a deep refiner network, alongside non-parametric transformations, in a one-time offline process to encode rich representations into a compact and efficient embedding structure. This approach decouples slow neural networks with high representational capacity from the fast inference path. We introduce DRR-Net, a simple network that validates this paradigm, and a novel data augmentation strategy, Variational Pairs (VP) for improving INRs under complex tasks like high-dimensional surrogate modeling. Experiments on several ensemble simulation datasets demonstrate that our approach achieves state-of-the-art fidelity, while being up to $27\times$ faster at inference than high-fidelity baselines and remaining competitive with the fastest models. The DRR paradigm offers an effective strategy for building powerful and practical neural field surrogates and INRs in broader applications, with a minimal compromise between speed and quality.

## 1 INTRODUCTION

Surrogate modeling of large-scale scientific simulations is a critical task in fields ranging from climate science to cosmology, enabling accelerated discovery and analysis. Neural Fields, such as Implicit Neural Representations (INRs), have emerged as a powerful framework for this task, offering a continuous and flexible coordinate-based representation. For individual or time-varying fields, INR-based methods have achieved remarkable reconstruction quality and inference speed (Takikawa et al., 2021; Müller et al., 2022; Weiss et al., 2022; Wurster et al., 2023; Fridovich-Keil et al., 2023; Wurster & Shen, 2025). However, a major challenge remains in scaling these models to higher-dimensional ensemble simulations, where collections of outcomes are generated under varying parameters. Existing INR-based surrogates for ensembles struggle to retain both high fidelity and computational efficiency (Li et al., 2025; Chen et al., 2025), presenting challenges to practical applications of INRs for ensemble simulation analysis that balance reconstruction quality and computation efficiency.

This fidelity-speed trade-off is rooted in the architectures of modern INRs: embedding-based versus MLP-based approaches. Embedding-based methods, which often utilize learnable feature grids,

---

[*]Code available at `https://github.com/xtyinzz/DRR-INR`

achieve high efficiency by encoding information into an explicit structure queried via fast interpolation (Fridovich-Keil et al., 2022; Müller et al., 2022). However, naively scaling these structures to represent both spatial coordinates and simulation parameters incurs prohibitive memory costs. A common workaround employs low-rank or decomposed representations, factorizing the spatial and conditional inputs into separate, low-dimensional structures (Chen et al., 2022; Fridovich-Keil et al., 2023; Chen et al., 2025). While memory-efficient, this factorization acts as a representational bottleneck, struggling to capture complex, non-linear interactions within the data. In contrast, MLP-based architectures can model high-dimensional functions directly, making them inherently suitable for capturing the intricate relationships in ensemble data. For instance, FA-INR (Li et al., 2025) leverages a Mixture of Experts (MoE) (Jacobs et al., 1991; Shazeer et al., 2017) of MLPs to achieve state-of-the-art fidelity. This expressive power, however, comes at a steep computational price: the sequential and heavyweight MLPs, even in a MoE setup, result in high inference latency, rendering large-scale data analysis and interactive exploratory workflows infeasible.

Towards resolving the trade-off between representation quality and inference efficiency, we introduce a new architectural paradigm: Decoupled Representation Refinement (DRR). The core insight is to enrich an INR with deep nonlinear mappings for representation refinement while decoupling the expensive network evaluation from inference. We achieve this with refiner networks and non-parametric transformations that operate directly on the embeddings within the INR's feature structures, rather than on dynamic inputs or intermediate activations, to encode more complex simulation features. Crucially, this refinement process can be precomputed and its results cached, reducing the amortized inference cost to merely that of embedding interpolation and a lightweight decoder.

To validate the efficacy of the DRR paradigm, we introduce DRR-Net, a simple architecture that implements the core principle, and demonstrate that this approach achieves state-of-the-art fidelity and speed against leading INR-based surrogates. Furthermore, training a generalizable INR to model complex field variations across the continuous spatio-conditional space presents a significant challenge, particularly given the prohibitive cost of simulation which results in sparse ensemble training data in both the spatial and simulation condition domains. To mitigate this data scarcity, we introduce Variational Pairs (VP), a versatile data augmentation strategy that provides robust performance gains across diverse models and datasets.

- We propose **Decoupled Representation Refinement (DRR)**, an architectural framework that enhances the expressive capacity of INRs without sacrificing inference efficiency.
- We design **DRR-Net**, a simple architectural implementation that achieves state-of-the-art fidelity and efficiency against INR-based surrogates, validating the efficacy of DRR.
- We introduce **Variational Pairs (VP)**, a general-purpose data augmentation for INR datasets that provides significant performance gains for diverse model architectures.
- We conduct a **comprehensive, multi-faceted evaluation**, elucidating the fidelity-speed properties of INR models on challenging downstream tasks, including conditional generalization and zero-shot spatio-conditional generalization on ensemble simulations, while further demonstrating DRR's versatility in computer vision and graphics applications.

## 2 RELATED WORK

**Implicit Neural Representation (INR) Architectures.** INR architectures have evolved from pure MLP-based models to more efficient hybrid approaches to balance fidelity and speed. While MLPs using Fourier feature encodings (Mildenhall et al., 2020; Tancik et al., 2020), or sinusoidal and wavelet activations (Sitzmann et al., 2020; Fathony et al., 2020; Saragadam et al., 2023; Liu et al., 2024), can achieve high fidelity, they remain computationally expensive. We refer to Essakine et al. (2024) for a more comprehensive review of state-of-the-art MLP-based models. Another paradigm instead leverages fast, explicit embedding structures, including dense grids (Martel et al., 2021; Fridovich-Keil et al., 2022; Weiss et al., 2022; Xiong et al., 2024), hierarchical octrees and hash grids (Takikawa et al., 2021; Müller et al., 2022), and decomposed structures (Chen et al., 2022; Chan et al., 2022; Fridovich-Keil et al., 2023; Essakine et al., 2024), paired with a small decoder MLP. Our DRR paradigm synthesizes the two philosophies with a decoupled refiner to enhance the quality of an efficient embedding structure, representing a step towards more practical INRs that retain both quality and speed at deployment.

**INR Acceleration Methods.** Orthogonal to the architectural approach for fast INRs, efficiency can also be realized through alternative methodologies. In the context of Neural Radiance Fields (NeRF) (Mildenhall et al., 2020), rendering-specific optimizations, such as efficient volume integral computation (Wu et al., 2022) and informed ray sampling (Kurz et al., 2022; Barron et al., 2022; Gupta et al., 2023), are widely studied. However, NeRF-specific algorithms are generally not applicable to data representation INRs. Conversely, general acceleration techniques such as learning separate parameters for decomposed domain partitions (Reiser et al., 2021; Saragadam et al., 2022; Wurster et al., 2023) or employing knowledge distillation (Reiser et al., 2021; Barron et al., 2022; Duckworth et al., 2024) are highly relevant. Crucially, these strategies are complementary to our architectural contributions; the DRR paradigm can serve as the efficient backbone model within these decomposed or distilled frameworks. Our work focuses on isolating the fundamental architectural perspective in delivering a fast and high-fidelity INR.

**Ensemble Simulation Surrogate Modeling.** Surrogate modeling accelerates scientific discovery by replacing costly simulations with efficient approximations to explore the parameter-outcome relationships (Forrester et al., 2008; Alizadeh et al., 2020). While various machine learning approaches are used for tasks like parameter space exploration and sensitivity analysis (Gramacy et al., 2004; Hazarika et al., 2019; Shi et al., 2022b;a), INR-based surrogates offer superior flexibility by representing the ensemble as a single continuous function that can be queried at arbitrary locations. However, these methods are bound by a clear fidelity-speed trade-off: high-fidelity, MLP-based models like FA-INR (Li et al., 2025) are computationally expensive, while fast, embedding-based models like Explorable-INR (Chen et al., 2025) struggle to match the accuracy.

## 3 METHODOLOGY

### 3.1 PRELIMINARIES: CONDITIONAL INRS FOR SURROGATE MODELING

Ensemble simulations are critical tools in science, allowing researchers to explore the complex relationship between experimental conditions and resulting phenomena, such as yeast cell polarization (Yi et al., 2007; Renardy et al., 2018) or ocean temperature and salinity dynamics (Shi et al., 2022b). We formally define the resulting dataset as follows. Let $\mathcal{P} \subset \mathbb{R}^{d_c}$ be the space of input condition parameters. A deterministic simulation, defined by a parameter vector $c \in \mathcal{P}$, yields an outcome field $\Phi_c$. This field is typically represented as a discrete set of $N$ coordinate-value pairs, $\Phi_c = \{(x_j, v_j)\}_{j=1}^N$, where each coordinate vector $x_j \in \mathbb{R}^{d_x}$ and $v_j \in \mathbb{R}^{d_v}$ is the corresponding field value. An ensemble dataset $\mathcal{D}$ is then a collection of $M$ such fields, generated from a discrete set of sampled parameters $\mathcal{C} = \{c_1, \ldots, c_M\} \subset \mathcal{P}$, such that $\mathcal{D} = \{(c_i, \Phi_{c_i})\}_{i=1}^M$.

INR-based surrogates are a powerful approach for this task (Chen et al., 2025; Li et al., 2025), as they learn a single, continuous function, $f_\theta$, that represents the entire ensemble. This enables flexible data exploration across the continuous domain of conditions $c \in \mathcal{P}$ and coordinates $x \in \mathbb{R}^{d_x}$. This function $f_\theta$ is typically realized using a conditional INR architecture, where the spatial coordinate $x$ and condition $c$ are encoded through a spatial encoder ($E_{sp}$) and a condition encoder ($E_{cond}$), respectively. Their feature outputs are then fused via a conditioning operation ($\circ$) and passed to a final lightweight decoder MLP ($g$). The entire mapping, with learnable parameters $\theta = \{\theta_{sp}, \theta_{cond}, \theta_{dec}\}$ optimized by minimizing the L1 or L2 loss between the network's predictions $f_\theta(x_j, c_i)$ and the ground truth values $v_j$, can be expressed as:

$$f_\theta(x, c) = g(E_{sp}(x; \theta_{sp}) \circ E_{cond}(c; \theta_{cond}); \theta_{dec}) \tag{1}$$

### 3.2 DECOUPLED REPRESENTATION REFINEMENT

A central challenge in designing INR-based surrogates is balancing reconstruction fidelity and inference speed, which are two critical metrics that determine the practical success of any simulation surrogate (Alizadeh et al., 2020). However, current state-of-the-art models fundamentally struggle to resolve this trade-off. The fundamental dilemma in designing INR-based surrogates is rooted in a core architectural choice: fast, embedding-based models often act as a representational bottleneck, while expressive, deep MLP-based models suffer from prohibitive inference latency. To resolve this, we propose the Decoupled Representation Refinement (DRR) paradigm, which is founded on the key insight that the expensive computation needed to build a high-capacity representation can be

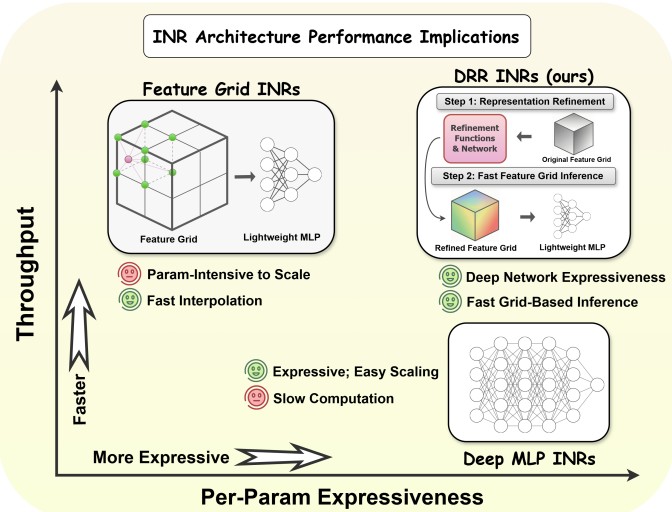

Figure 1: Our Decoupled Representation Refinement (DRR) paradigm synthesizes the strengths of slow, high-fidelity MLPs and fast, less expressive feature grid, or embedding-based models (compared at similar model sizes) as a well-rounded design pattern for fast and accurate INRs.

separated from the fast query path required at inference. We begin by detailing the formulation of DRR, followed by an analysis of its benefits.

### 3.2.1 DRR FORMULATION

Formally, we start with a baseline embedding-based INR defined by its learnable structure $\mathcal{G}$ and a query function $I$ that produces a feature $z = I(x; \mathcal{G})$. The DRR paradigm enhances this in three conceptual steps. First, to create a richer representation for refinement, we can apply a parameter-free transformation $\pi$ to the base structure $\mathcal{G}$, yielding an intermediate representation, $\hat{\mathcal{G}} = \pi(\mathcal{G})$. Second, and most critically, we can introduce a deep, non-linear refiner network, $R_\psi$. The refiner's purpose is to learn a powerful, non-linear transformation of the preprocessed embeddings. We denote this learned transformation as a refinement offset, $\Delta\mathcal{G}$:

$$\Delta\mathcal{G} = R_\psi(\hat{\mathcal{G}}) \tag{2}$$

Finally, the complete representation, $\mathcal{G}'$, is composed via a residual connection. This step sums the learned offset with the preprocessed base, synergistically combining the complementary representations from both:

$$\mathcal{G}' = \hat{\mathcal{G}} + \Delta\mathcal{G} = \pi(\mathcal{G}) + R_\psi(\pi(\mathcal{G})) \tag{3}$$

The final feature $z_{DRR}$ is then extracted from this cached structure using the same efficient query function: $z_{DRR} = I(x; \mathcal{G}')$. Thus, DRR provides an architectural framework for incorporating the expressiveness of a deep neural transformation into a computationally efficient embedding structure. During training, all parameters, including the base structure $\mathcal{G}$ and the refiner $R_\psi$, are jointly optimized to minimize the L2 loss for points in fields for training. We now analyze the key benefits that arise from this design.

### 3.2.2 IMPLICATIONS OF DRR

The simplicity of the DRR paradigm yields a set of architectural properties that resolve the conventional fidelity-speed trade-off. We analyze the three primary implications of this design: first, how refinement-decoupled inference achieves the speed of embedding-based models; second, how the framework promotes synergistic representation learning to improve expressiveness; and finally, how the preprocessing stage allows for further representation enhancement.

**Refinement-Decoupled Inference.** The foremost advantage of the DRR paradigm is its ability to deliver the expressivity of a deep network while matching the inference efficiency of purely embedding-based models. During inference, the deep refiner network $R_\psi$ is evaluated only once

in a pre-computation step to produce and cache the static, refined structure $\mathcal{G}'$ with Eq. 3. After this one-time cost is paid, the refiner is discarded. Consequently, all subsequent queries operate solely on the cached structure, reducing the amortized computational cost of inference to that of a standard embedding-based INR: a simple interpolation and a lightweight decoder.

**Synergistic Representation Learning.** DRR enhances expressivity by resolving the "entangled representation" problem. In standard INRs, embeddings must simultaneously capture global structure and local detail. The refiner decouples this by learning a shared decoding function across the grid, freeing the base embeddings to specialize in encoding dense, latent high-frequency signals. This creates a tightly coupled system: the base acts as a specialized pre-conditioner, optimizing for the refiner to suppress aliasing and decode physical quantities, rather than functioning as a standalone approximation (see Sec. G).

Second, the refiner acts as an offline fusion module. While common embedding-based INRs burden the lightweight MLP decoder with fusing multi-scale features (Müller et al., 2022; Weiss et al., 2022; Wurster et al., 2023), DRR offloads this complexity by learning to fuse features offline. By baking these refined representations directly into the cached grid $\mathcal{G}'$, we allow the online decoder to remain lightweight while operating on rich, pre-fused features.

**Embedding Preprocessing with Non-Parametric Refinements.** To ensure the proper optimization of the deep refiner network, the richness of the input information and the quantity of data points are critical factors. Since the refiner operates directly on the feature structures, these requirements correspond to the embedding dimensionality and the grid resolution, respectively. Satisfying these needs in a standard setup would necessitate base grids with high resolutions and large embedding dimensions, but such a design introduces a prohibitive parameter overhead. To resolve this trade-off and facilitate refiner optimization, we apply non-parametric transformations to synthesize a high-capacity feature manifold without increasing storage costs. We leverage two strategies. First, *Structural Super-Resolution* deterministically upsamples the grid to increase feature density. Second, *feature upsampling* with Positional Encoding (Mildenhall et al., 2020) projects features into a higher-dimensional space to capture complex frequencies. As validated in Sec. F, these operations serve as critical enablers that allow the refiner to generalize effectively while maintaining a compact base structure.

### 3.3    INSTANTIATING THE DRR PARADIGM WITH DRR-NET

To validate our paradigm, we introduce DRR-Net, a concrete and straightforward instantiation, as illustrated in Fig. 2, designed to demonstrate how the general DRR formulation can be adapted to create a powerful surrogate for ensemble simulations.

**The Multi-Resolution Unification Principle**. Modern, high-performance embedding structures frequently rely on multi-resolution representations (Müller et al., 2022; Wurster et al., 2023). To apply DRR to such structures, we introduce a general unification principle. The core challenge is preparing a single, unified input for the refiner from a set of embeddings $\{\mathcal{G}^l\}_{l=1}^{L}$ at different resolutions. Our solution is a preprocessing function, $\pi_{\text{unify}}$, that first upsamples each structure $\mathcal{G}^l$ to a common resolution via an operator $\mathcal{U}^l$, the structural super-resolution step, and then concatenates their features channel-wise ($\|$):

$$\hat{\mathcal{G}}_{\text{unified}} = \pi_{\text{unify}}(\{\mathcal{G}^l\}_{l=1}^{L}) = \Big\|_{l=1}^{L} \mathcal{U}^l(\mathcal{G}^l) \tag{4}$$

This unification is a crucial mechanism that allows a deep refiner to learn from and enhance the complex, multi-scale representations from the modern embedding structures.

**Application to DRR-Net Encoders.** The DRR-Net architecture, shown in Fig. 2, applies this unification principle to two distinct branches: a spatial encoder and a condition encoder.

The *Spatial Encoder*, depicted in Fig. 2(b), begins with a set of $L_{sp}$ multi-resolution feature grids, $\{\mathcal{G}_{sp}^l\}_{l=1}^{L_{sp}}$. To be explicit, each of the $L_{sp}$ grids has its own spatial resolution but shares a common feature dimension of $d_{fs}$ at each vertex. As the first step of unification, each grid $\mathcal{G}_{sp}^l$ is upsampled to a common, high resolution via structural super-resolution using interpolation. Then, as shown in Eq. 4, upsampled features from all $L_{sp}$ grids are concatenated channel-wise at each vertex. This process produces a single, dense, unified grid, $\hat{\mathcal{G}}_{unified}$ with vertex embeddings of dimension $L \times d_{fs}$.

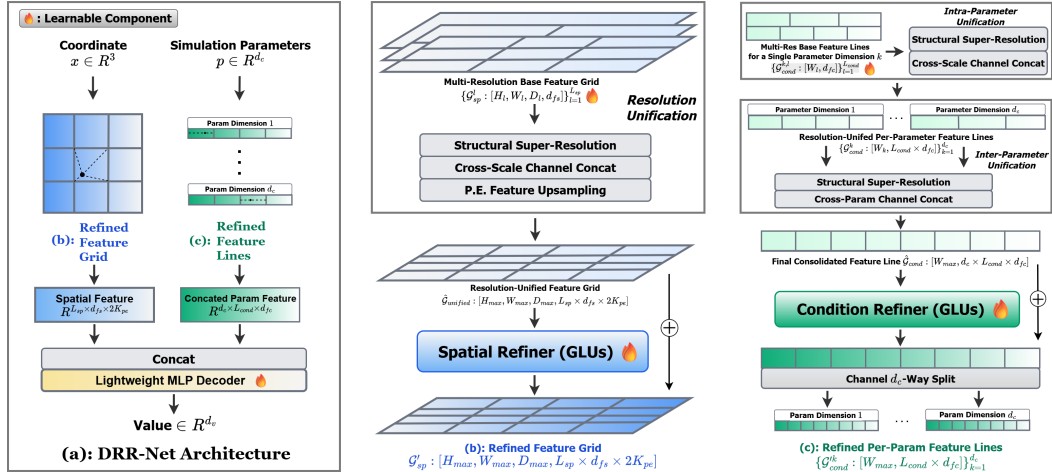

Figure 2: Concrete instantiation of the DRR paradigm via DRR-Net. Part (a) illustrates the overall architectural components, while part (b) and (c) detail the unification and refinement procedure applied to the base multi-resolution embedding structures.

Optionally, P.E. feature upsampling (with $K_{pe}$ frequencies) can then be applied to this unified representation, lifting its internal dimension to $L_{sp} \times d_{fs} \times 2K_{pe}$. The resulting grid serves as the input to the spatial refiner, which produces the final refined grid $\mathcal{G}'_{sp}$. To encode an input coordinate $x$, it is interpolated from this refined grid $\mathcal{G}'_{sp}$, producing a single embedding of the final dimensionality.

The ***Condition Encoder***, depicted in Fig. 2(c), is designed to be scalable to an arbitrary number of simulation parameters. For a simulation with $d_c$ parameters, the encoder initializes $d_c$ sets of $L_{cond}$ multi-resolution 1D feature lines, $\{\mathcal{G}_{cond}^{k,l}\}$, each with a base feature dimension of $d_{fc}$. These structures are unified via a hierarchical, "local-then-global" process to learn crucial cross-parameter feature fusion lacking in the low-rank setup. First, an intra-parameter (local) unification step is applied to the $L_{cond}$ lines for each parameter separately. It upsamples each parameter's lines to their own common (highest) resolution and concatenates their features, producing $d_c$ independent, unified feature lines, $\{\mathcal{G}_{cond}^{k}\}_{k=1}^{d_c}$. Each of these lines now has a feature dimension of $L_{cond} \times d_{fc}$. Second, an inter-parameter (global) unification step unifies these $d_c$ lines across all parameters. As shown in Eq. 5, this step upsamples all $d_c$ lines to a single, global high resolution (via $\mathcal{U}^k$) and concatenates them channel-wise.

$$\hat{\mathcal{G}}_{cond} = \pi_{unify}(\{\mathcal{G}_{cond}^k\}_{k=1}^{d_c}) = ||_{k=1}^{d_c} \mathcal{U}^k(\mathcal{G}_{cond}^k) \tag{5}$$

This results in the final, consolidated 1D representation, $\hat{\mathcal{G}}_{cond}$, which is then processed by the condition refiner. Crucially, after refinement, this single line is split back into $d_c$ dedicated feature lines, one for each parameter, which now contain richly fused cross-resolution and cross-parameter features. To encode the $d_c$-dimensional input parameters, we interpolate from each of the refined lines, yielding $d_c$ feature vectors. Since the refiner's output dimension matches its input as with the residual formulation, each vector has a size of $L_{cond} \times d_{fc}$. These are then concatenated to produce the final conditional feature for the decoder, with a total dimension of $d_c \times L_{cond} \times d_{fc}$.

## 3.4 DATA AUGMENTATION WITH VARIATIONAL PAIRS

Training INR surrogates on a sparse set of ensemble members is challenging, making data augmentation a crucial tool for improving generalization. A related technique, Variational Coordinates (VC) (Zhang et al., 2024), was introduced to perturb input coordinates for anti-aliasing. However, we find that naively applying VC as an augmentation strategy can be ineffective or even harm model accuracy. We hypothesize this failure stems from a substantial mismatch between the generated data and the ground truth. Specifically, while the true function that maps coordinates to values in a scientific simulation is often highly complex and continuous, the VC augmentation implicitly enforces a piecewise-constant assumption by keeping values unchanged for perturbed coordinates. Our core

insight is to resolve this mismatch. We propose a new family of augmentations, Variational Pairs (VP), that generates a new, physically-plausible value corresponding to the perturbed coordinate.

**VP-S: Spatial Augmentation.** Our first strategy, VP-S, is a general-purpose augmentation applicable to any INR. It generates a perturbed coordinate $\tilde{x} = x + \epsilon_x$, where the perturbation $\epsilon_x$ is sampled from a zero-mean isotropic Gaussian distribution, $\epsilon_x \sim \mathcal{N}(0, \sigma^2 \mathbf{I})$, and is truncated at a distance threshold $\tau$. This follows the formulation of VC (Zhang et al., 2024). Crucially, we approximate the new value $\tilde{v}$ by interpolating the original field $\Phi_c$ at this new location. This interpolation function, $I(\Phi_c, \tilde{x})$, is a weighted average of the ground truth values from a local set of neighboring vertices, $\mathcal{V}(\tilde{x})$, on the original simulation grid:

$$\tilde{v} = I(\Phi_c, \tilde{x}) = \sum_{j \in \mathcal{V}(\tilde{x})} w_j(\tilde{x}) v_j \tag{6}$$

where the set of neighbors $\mathcal{V}(\tilde{x})$ and their corresponding weights $w_j(\tilde{x})$ are determined by the chosen interpolation function. While any interpolation imposes its own prior, this assumption of local smoothness is much milder and more physically plausible than the piecewise-constant one. The resulting pair $(\tilde{x}, \tilde{v})$ therefore provides a higher-quality supervisory signal for learning a locally smooth and better representation of the field than VC.

**VP-SC: Spatio-Conditional Augmentation.** For the specific challenge of conditional INR tasks like ensemble surrogates, we propose a more integrated strategy, VP-SC, that augments both spatial and conditional inputs. Given a sample from a field $\Phi_{c_i}$, we generate a perturbed query $(\tilde{x}, \tilde{c})$ by adding truncated Gaussian noise to both the coordinate and the condition parameter. The key challenge is to approximate the unknown value $\tilde{v}$ for this novel condition $\tilde{c}$. We estimate it via a two-stage interpolation scheme. First, we identify the $K$-nearest neighbor conditions $\{c_k\}_{k=1}^{K}$ to the original condition $c_i$ from the training set $\mathcal{C}$. We perform a spatial interpolation for the perturbed coordinate $\tilde{x}$ within each of these $K$ neighbor fields using Eq. 6, yielding a set of candidate values $\{v'_k = I(\Phi_{c_k}, \tilde{x})\}_{k=1}^{K}$. Second, we perform a conditional interpolation on these candidate values using Inverse Distance Weighting (IDW) to compute the final value:

$$\tilde{v} = \sum_{k=1}^{K} w_k(\tilde{c}) v'_k, \quad \text{where} \quad w_k(\tilde{c}) = \frac{1/\|\tilde{c} - c_k\|_2}{\sum_{j=1}^{K} 1/\|\tilde{c} - c_j\|_2} \tag{7}$$

This process is designed to generate plausible training samples in the continuous spatio-conditional space, providing the model with a more diverse data distribution than the original sparse samples. The goal is to encourage the model to learn the underlying ensemble dynamics more effectively and improve its generalization to unseen conditions.

## 4 EXPERIMENTS

We conduct a series of experiments to rigorously evaluate our proposed DRR paradigm and DRR-Net architecture. First, we benchmark DRR-Net against state-of-the-art baselines on the primary task of conditional generalization, assessing both fidelity and efficiency (Sec. 4.2 with additional visualizations in Sec. C). Second, we introduce a more challenging test of zero-shot spatio-conditional generalization to probe the quality of the learned continuous representation (Sec. 4.3). Finally, we present an ablation study over the data augmentation strategies to evaluate our proposed VP methods (Sec. 4.4). A more comprehensive suite of DRR components, including the refiners $R_\psi$, preprocessing functions $\pi$, and VP hyperparameters are provided in the appendix (Sec. E and beyond).

To demonstrate the versatility of the DRR paradigm beyond scientific surrogates, we also present its application to broader vision and graphics tasks in Sec. I.

### 4.1 EXPERIMENTAL SETUP

**Datasets.** We evaluate our method on three 3D ensemble simulation datasets, consistent with prior work in INR-based surrogate modeling (Chen et al., 2025; Li et al., 2025). These include the cosmological simulation Nyx (Almgren et al., 2013), the oceanography simulation MPAS-Ocean (Ringler et al., 2013), and the hydrodynamics simulation Cloverleaf3D (Mallinson et al., 2013). Key statistics for each dataset, including the number of ensemble members, parameter dimensions, and spatial resolution, are summarized in Tab. 1 with more details in Sec. B.4.

Table 1: Description of ensemble simulation datasets used for evaluations.

| Dataset | Structure | Variable | # Parameters | Resolution | # Training | # Testing |
|---|---|---|---|---|---|---|
| Nyx | 3D Cartesian Grid | Dark Matter Density | 3 | $256^3$ | 100 | 30 |
| MPAS-Ocean | Voronoi Mesh | Ocean Temperature | 4 | 11845146 vertices | 70 | 30 |
| Cloverleaf3D | 3D Cartesian Grid | Energy | 6 | $128^3$ | 500 | 100 |

**Baselines.** We compare DRR-Net against three state-of-the-art methods designed for high-dimensional fields and ensemble simulations, which represent the two main INR architectural approaches. For efficient, embedding-based models, we compare against (1) K-Planes (Fridovich-Keil et al., 2023), which uses decomposed 2D feature planes, and (2) Explorable-INR (Chen et al., 2025), which uses a factorized grid representation. To represent high-fidelity, MLP-based models, we compare our approach against (3) FA-INR (Li et al., 2025), a concurrent work that leverages a mixture of MLP experts for spatial feature encoding with cross-attention-based conditioning.

**Metrics.** We evaluate all models on both computational efficiency and prediction fidelity to assess their practicality as simulation surrogates. For efficiency, we measure four key indicators: the computational cost of inference (TFLOPs per $10^9$ points) to assess theoretical complexity; the total inference time (in seconds) to simulate data analysis latency; the total training time (in hours); and the model size (number of parameters). For fidelity, we report the Relative L2 error (Rel L2), Peak Signal-to-Noise Ratio (PSNR), and the Structural Similarity Index Measure (SSIM) (Wang et al., 2004). As the standard SSIM computation relies on a sliding window over a regular grid, we do not report it for the MPAS-Ocean dataset with an unstructured mesh topology.

Table 2: Results for the conditional generalization task. DRR-Net achieves a state-of-the-art fidelity and efficiency in all datasets, with more pronounced benefits for Nyx and Cloverleaf3D.

| Dataset | Model | Rel L2↓ | PSNR↑ | SSIM↑ | TFLOPs / $10^9$ pts ↓ | Inference Time (sec) | Training Time (hr) | # Params |
|---|---|---|---|---|---|---|---|---|
| Nyx | K-Planes | 1.96e-01 | 28.86 | 0.797 | 57.0 | 21.6s | **23.8h** | 12.1M |
| | FA-INR | 3.95e-02 | 42.79 | 0.975 | 2569.2 | 287.2s | 64.1h | 9.5M |
| | Explorable-INR | 4.64e-02 | 41.39 | 0.972 | **39.6** | **9.6s** | 24.7h | 14.7M |
| | DRR-Net (ours) | **3.18e-02** | **44.69** | **0.986** | 57.2 | 10.7s | 24.0h | **8.9M** |
| MPAS-Ocean | K-Planes | 3.81e-01 | 15.45 | - | 60.0 | 16.9s | 6.6h | 13.5M |
| | FA-INR | **5.58e-03** | **52.13** | - | 422.1 | 89.8s | 21.1h | **1.0M** |
| | Explorable-INR | 1.13e-02 | 46.03 | - | **39.6** | 5.3s | **3.3h** | 14.7M |
| | DRR-Net (ours) | 7.76e-03 | 49.26 | - | 47.3 | **4.8s** | 7.2h | 4.6M |
| Cloverleaf3D | K-Planes | 8.42e-01 | 30.02 | 0.787 | 67.7 | 17.7s | **26.7h** | 6.8M |
| | FA-INR | 1.11e-01 | 47.60 | 0.991 | 422.1 | 56.2s | 29.4h | 1.0M |
| | Explorable-INR | 1.46e-01 | 45.22 | 0.984 | **39.6** | 5.3s | 28.0h | 4.5M |
| | DRR-Net (ours) | **9.81e-02** | **48.69** | **0.994** | 52.6 | 5.5s | 44.0h | **0.9M** |

## 4.2 CONDITIONAL GENERALIZATION EVALUATION

Our first experiment evaluates DRR-Net on the primary task of conditional generalization, assessing its ability to predict entire fields for unseen simulation parameters. The results, presented in Tab. 2, demonstrate that DRR-Net successfully overcomes the conventional trade-off between fidelity and efficiency. On both the Nyx and Cloverleaf3D datasets, DRR-Net achieves state-of-the-art reconstruction quality while maintaining inference speeds competitive with the fastest embedding-based methods. For example, on Nyx, DRR-Net obtains the highest PSNR (44.69), yet is over $27\times$ faster during inference than the high-fidelity FA-INR. Furthermore, DRR-Net is highly parameter-efficient, achieving these results with a smaller model size than all other embedding-based baselines.

On the MPAS-Ocean dataset, we observe that the MLP-based FA-INR achieves a higher PSNR. We attribute this to a structural mismatch between the dataset's underlying geometry and the inductive bias of grid-based methods. MPAS-Ocean is simulated on an unstructured spherical mesh with an adaptive vertex density, whereas feature grids, as used in our and other embedding-based models, interpolate in a Cartesian system assuming uniform grid spacing. An MLP, lacking a spatial inductive bias, is less affected by this mismatch. Crucially, however, DRR-Net still significantly outperforms the other grid-based method, Explorable-INR, by 3 dB in PSNR. This provides strong evidence that

our DRR paradigm is effectively compensating for the inherent limitations of the base grid representation, demonstrating its robustness. Additionally, we demonstrate through visualization that our DRR-Net can qualitatively capture temperature features as well as FA-INR in Sec. C.3 while being substantially faster.

Table 3: Spatio-conditional generalization results. DRR-Net consistently attains the best fidelity with highly competitive inference efficiency matching the embedding-only model Explorable-INR.

| Dataset | Model | Unseen Fields (Rel L2↓ / PSNR↑ / SSIM↑) | Trained Fields (Rel L2↓ / PSNR↑ / SSIM↑) | Inference Time (sec) | # Params |
|---|---|---|---|---|---|
| Nyx | K-Planes | 1.89e-01 / 29.18 / 0.709 | 2.91e-01 / 29.62 / 0.716 | 75.3s | **3.9M** |
| | FA-INR | 4.55e-02 / 41.57 / 0.971 | 7.29e-02 / 41.65 / 0.971 | 1245.1s | 9.5M |
| | Explorable-INR | 6.48e-02 / 38.50 / 0.959 | 8.44e-02 / 40.38 / 0.963 | **38.6s** | 10.0M |
| | DRR-Net (ours) | **4.31e-02 / 42.04 / 0.975** | **7.07e-02 / 41.92 / 0.975** | 43.7s | 8.7M |
| Cloverleaf3D | K-Planes | 4.47e-01 / 35.52 / 0.914 | 5.78e-01 / 36.04 / 0.919 | 81.5s | 3.1M |
| | FA-INR | 1.07e-01 / 47.98 / 0.992 | 9.15e-02 / 52.06 / **0.996** | 327.8s | 1.0M |
| | Explorable-INR | 1.25e-01 / 46.58 / 0.989 | 1.10e-01 / 50.49 / 0.994 | 27.3s | 4.0M |
| | DRR-Net (ours) | **1.01e-01 / 48.43 / 0.993** | **8.70e-02 / 52.49 / 0.996** | **25.6s** | **0.9M** |

## 4.3 SPATIO-CONDITIONAL GENERALIZATION EVALUATION

Having established DRR-Net's performance on the standard task, we now probe the fundamental quality of INRs with a more challenging test of spatio-conditional generalization. To do this, we evaluate all models on zero-shot super-resolution for both trained and unseen ensemble members. Models are trained on fields that have been downsampled to $\times 2$ lower resolutions and are then evaluated on their ability to reconstruct the original, full-resolution fields. For example, Nyx training fields are downsampled to $128^3$ and evaluated at $256^3$. This task is crucial as it assesses the model's ability to learn a truly continuous representation across both spatial and conditional domains. We also evaluate the condition-only generalization directly on the downsampled resolution in Sec. D to study the fidelity and efficiency of models in smaller-scale problems.

The results, presented in Tab. 3, demonstrate that DRR-Net's superior performance holds. For both unseen and trained fields, we report the relative L2 error, PSNR, and SSIM in a single cell. On both the Nyx and Cloverleaf3D datasets, DRR-Net achieves the highest reconstruction quality for both trained and test simulation parameter settings at the unseen full resolution, surpassing the highest-fidelity baseline FA-INR while being 11-28$\times$ faster to generate the fields. It shows that the benefits of our DRR paradigm generalize robustly across both spatial scales and condition parameters, demonstrating the efficacy of DRR-Net as a continuous implicit representation.

## 4.4 ABLATION STUDY: DATA AUGMENTATION

We conduct an ablation study to validate our proposed Variational Pairs (VP) data augmentation. We compare our two variants, VP-S and VP-SC, against a no-augmentation baseline and the Variational Coordinate (VC) method (Zhang et al., 2024). The analysis, presented in Tab. 4, is performed across all baseline models on our two primary Cartesian grid simulations, Nyx and Cloverleaf3D.

First, the results confirm the risks of the naive VC approach. Its performance is highly unpredictable; while it sometimes offers a marginal benefit, it is detrimental to DRR-Net on both datasets, and this effect is more pronounced for Explorable-INR in Cloverleaf3D. This supports our reasoning that the implicit piecewise-constant assumption provides a low-quality supervisory signal.

In contrast, our proposed VP methods provide consistent and substantial performance gains across the vast majority of model-dataset combinations. The simpler VP-S strategy emerges as a remarkably robust and effective all-arounder. On Cloverleaf3D, VP-S is the top-performing augmentation for three of the four models, including our own DRR-Net (+1.79 dB). The more complex VP-SC also proves effective, delivering the best performance for the baselines on Nyx, though its additional complexity does not always translate to superior gains over VP-S. This suggests a trade-off between the methods. We note that the efficacy of VP methods can be further influenced by hyperparameter tuning, as we detail in Appendix H.

Table 4: Ablation study of data augmentation strategies, showing that proposed VP methods consistently outperform the no-augmentation and VC baselines across diverse INR architectures.

| Model | Augment Method | Nyx | | | Cloverleaf3D | | |
|---|---|---|---|---|---|---|---|
| | | Rel L2↓ | PSNR↑ | SSIM↑ | Rel L2↓ | PSNR↑ | SSIM↑ |
| K-Planes | None | 2.07e-01 | 28.41 | 0.783 | 8.57e-01 | 29.87 | 0.784 |
| | VC | 1.98e-01 | 28.80 | 0.792 | 8.87e-01 | 29.57 | 0.775 |
| | VP-S (ours) | 2.05e-01 | 28.48 | **0.804** | **7.89e-01** | **30.59** | **0.801** |
| | VP-SC (ours) | **1.96e-01** | **28.86** | 0.797 | 8.42e-01 | 30.02 | 0.787 |
| FA-INR | None | 4.96e-02 | 40.81 | 0.961 | 6.40e-01 | 32.40 | 0.764 |
| | VC | 4.91e-02 | 40.91 | 0.963 | 1.01e-01 | 48.47 | 0.992 |
| | VP-S (ours) | 4.31e-02 | 42.04 | 0.970 | **9.33e-02** | **49.13** | **0.993** |
| | VP-SC (ours) | **3.95e-02** | **42.79** | **0.975** | 1.11e-01 | 47.60 | 0.991 |
| Explorable-INR | None | 8.14e-02 | 36.51 | 0.932 | 1.50e-01 | 45.03 | **0.984** |
| | VC | 5.71e-02 | 39.59 | 0.964 | 2.02e-01 | 42.41 | 0.980 |
| | VP-S (ours) | 4.69e-02 | 41.30 | 0.971 | 1.67e-01 | 44.07 | 0.978 |
| | VP-SC (ours) | **4.64e-02** | **41.39** | **0.972** | **1.46e-01** | **45.22** | **0.984** |
| DRR-Net (ours) | None | **3.09e-02** | **44.92** | **0.986** | 1.13e-01 | 47.47 | 0.993 |
| | VC | 3.81e-02 | 43.11 | 0.982 | 1.15e-01 | 47.29 | 0.992 |
| | VP-S (ours) | 3.20e-02 | 44.61 | 0.985 | **9.19e-02** | **49.26** | **0.994** |
| | VP-SC (ours) | 3.18e-02 | 44.69 | **0.986** | 9.81e-02 | 48.69 | **0.994** |

## 5 LIMITATIONS AND FUTURE WORK

While our work demonstrates the effectiveness of the DRR paradigm, we identify several limitations and promising avenues for future research that build upon our findings.

**Application to Embedding Structure Variants.** Our implementation of DRR-Net is limited to multi-resolution Cartesian feature grids. However, the DRR paradigm itself is agnostic to the specific embedding structure. A compelling direction for future work is to apply DRR to other advanced representations, such as hash grids (Müller et al., 2022), adaptive grids (Wurster et al., 2023) for scientific simulation, or other learned primitives like 3DGS (Kerbl et al., 2023).

**Alternative Refiner Architectures.** In this work, we only used a simple, point-wise GLU-based refiner to isolate the benefits of the DRR framework and establish a baseline. A significant opportunity exists in exploring more specialized refiner architectures. For instance, a CNN-based refiner could explicitly leverage the spatial locality of the embedding grids, while a Transformer-based refiner could model global, long-range dependencies between embedding features. Such architectures, guided by the DRR principle, could potentially unlock further performance improvements.

**Scope of Generalization.** Current INR-based ensemble surrogates, including this work, primarily target conditional generalization within the convex hull of the training parameters or interpolation. This scope aligns with standard surrogate modeling workflows where the parameter space of interest is pre-defined for exploration. However, extending INR surrogates to perform robust extrapolation, or predicting simulation behaviors outside the training range, presents a distinct challenge, particularly for data-driven approaches in modeling such complex non-linear dynamics. We identify this as a promising direction for future research to further enhance the utility and generality of neural field surrogates.

## 6 CONCLUSION

In this work, we addressed the critical trade-off between fidelity and efficiency in INR-based surrogates for large-scale ensemble simulations. We introduced the Decoupled Representation Refinement (DRR) paradigm for INR, a novel framework that leverages a deep refiner network in an offline stage to enhance the expressive power of a compact and efficient embedding structure. This approach successfully decouples the high representational capacity from the fast inference path. Our comprehensive experiments demonstrate that our implementation, DRR-Net, achieves state-of-the-art reconstruction quality on several scientific benchmarks while remaining efficient, similar to the fastest embedding-based methods. We believe that the DRR paradigm offers a general and effective strategy towards building the next generation of powerful and practical implicit neural field surrogates, paving the way for more efficient scientific data analysis and discovery.

ACKNOWLEDGMENTS

We gratefully acknowledge the financial support provided by the U.S. Department of Energy (DOE) SciDAC program under grants DE-SC0021360 and DE-SC0023193. Additionally, this research was supported by the National Science Foundation (NSF) Division of Information and Intelligent Systems (Grant IIS-1955764) and by Los Alamos National Laboratory under Contract C3435.

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

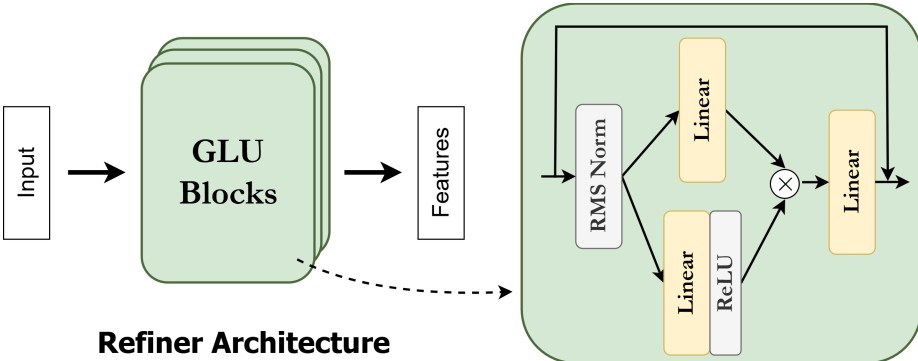

Figure 3: Architecture of the refiner network, which consists of stacked GLU blocks with pre-normalization and a ReLU intermediate activation.

Tau-Mu Yi, Shanqin Chen, Ching-Shan Chou, and Qing Nie. Modeling yeast cell polarization induced by pheromone gradients. *Journal of Statistical Physics*, 128(1):193–207, 2007.

Biao Zhang and Rico Sennrich. Root mean square layer normalization. *Advances in Neural Information Processing Systems*, 32, 2019.

Richard Zhang, Phillip Isola, Alexei A Efros, Eli Shechtman, and Oliver Wang. The unreasonable effectiveness of deep features as a perceptual metric. In *CVPR*, 2018.

Shuyi Zhang, Ke Liu, Jingjun Gu, Xiaoxu Cai, Zhihua Wang, Jiajun Bu, and Haishuai Wang. Attention beats linear for fast implicit neural representation generation. In *European Conference on Computer Vision*, pp. 1–18. Springer, 2024.

## A  LLM USAGE ACKNOWLEDGEMENT

We acknowledge using a large language model (LLM) to improve the final manuscript by correcting grammar, addressing typographical errors, and consolidating phrasing for conciseness and clarity. The authors take full responsibility for all content.

## B  IMPLEMENTATION DETAILS

Our source code is publicly available at `https://github.com/xtyinzz/DRR-INR`. This section provides further implementation details for the experiments presented in Sec. 4.

### B.1  REFINER GLU ARCHITECTURE

As described in Sec. 3.3, both refiners from the spatial and condition branches are composed of GLUs. We illustrate the detailed architecture in Fig. 3. Within each GLU block, the input is first processed through a pre-normalization layer using Root Mean Square Normalization (RMS Norm) (Zhang & Sennrich, 2019). The normalized output is then branched into two parallel paths containing separate linear transformations, and ReLU is applied to one path as a ReGLU formulation (Shazeer, 2020). The two sets of features are then multiplied, followed by a residual connection from the unnormalized input.

This ReGLU is a representative modern architecture for point-wise transformation, and the additional normalization and residual connections ensures both the refiner and the embeddings can receive proper level of gradients, which can be critical since they are among the earliest components of a DRR network, prone to vanishing gradient problems.

## B.2 SOFTWARE AND HARDWARE

All of our models and the baselines are implemented in Python using PyTorch. Crucially, no customized computing kernels or hardware-specific optimizations were used for any model. This ensures that all performance comparisons are based on architectural and algorithmic efficiency rather than low-level code optimization.

All experiments were conducted on a single node of a high-performance computing (HPC) cluster. Each experimental run in Sec. 4 was allocated 10 CPU cores (AMD EPYC 7H12), 128 GB of system RAM, and a single NVIDIA A100 GPU with 40GB of VRAM.

## B.3 EVALUATION METRICS

Because computational efficiency is one of the primary motivations for building surrogates (Alizadeh et al., 2020), we reported a suite of metrics over Sec. 4. This section provides the detailed procedures used to calculate these efficiency metrics.

**FLOPs.** We report the theoretical computational cost in TFLOPs (trillions of floating-point operations) per $10^9$ queried points. We chose this unit to approximate the cost of typical large-scale analysis tasks, such as reconstructing a single high-resolution $1024^3$ field or exploring a parameter space of 500 members at $128^3$ resolution.

To obtain a stable value, we first compute the average FLOPs for a forward pass on a representative batch of $10^2$ conditioning parameters and $10^3$ coordinates over 101 runs. We then compute a stable average and scale by $10^4$ to get the final reported number for $10^9$ points.

**Inference Time.** We report the wall-clock time required to reconstruct the full test set for each task. For the conditional generalization task (Tab. 2), this is the time to evaluate all unseen ensemble members. For the spatio-conditional task (Tab. 3), this is the time to generate the full-resolution fields for all members, since none were seen at high resolution during training.

**Training Time.** Measuring the exact end-to-end training duration can be inconsistent on HPC systems due to job scheduling. To ensure a fair comparison, we report a stable lower-bound estimate: we measure the average per-iteration time over 100 steps (once training is stable) and multiply it by the total number of training iterations. This serves as a consistent lower-bound estimate of the training time, excluding initializations, data-loading, and other overheads.

## B.4 DETAILS OF SIMULATION PARAMETERS FOR DATASETS

In this section, we provide a detailed description of the three 3D ensemble simulation datasets evaluated in this work. For all datasets, each ensemble member represents a transient simulation evolved over a specific time duration, with the scalar field of interest extracted from the final timestep. Across the ensemble, all simulation conditions remain constant except for the specific parameters selected for exploration. We detail the dataset-specific configurations below. For train/test splits and statistical overviews, please refer to Tab. 1 in the main text. Additionally, we provide the exact parameter value combinations for all training and testing members in the `dataset/param_source` directory of the open-sourced code.

**Nyx (Almgren et al., 2013)** is a cosmological hydrodynamics simulation developed by Lawrence Berkeley National Laboratory. The ensemble varies three cosmological parameters of interest: the total matter density ($OmM \in [0.12, 0.155]$), the baryon density ($OmB \in [0.0215, 0.0235]$), and the Hubble Constant ($h \in [0.55, 0.85]$). A total of 130 ensemble members were generated via random sampling within these parameter ranges. Each simulation was evolved for 200 timesteps.

**MPAS-Ocean (Ringler et al., 2013)** is an unstructured-mesh ocean model developed by Los Alamos National Laboratory. The ensemble explores the sensitivity to four parameters: the Bulk wind stress Amplification ($BwsA \in [0.0, 5.0]$), the Critical bulk Richardson number ($CbrN \in [0.25, 1.00]$), the Gent-McWilliams Mesoscale eddy transport parameterization ($GM \in [600.0, 1500.0]$), and the horizontal viscosity ($HV \in [100.0, 300.0]$). The simulation utilizes a spherical coordinate system with a Voronoi tessellation for the horizontal grids at each depth level. The dataset consists of 100 members, each simulated for a duration of 15 model days.

Table 5: Architectural hyperparameters for INR surrogate models across the evaluated ensemble simulation datasets.

| Model | Hyperparameter | Nyx | MPAS-Ocean | Cloverleaf3D |
|---|---|---|---|---|
| K-Planes | Spatial Plane Base Resolution | 32 | 32 | 16 |
| | Spatial Plane Multi-Res Multipliers | $[1, 2, 4, 8]$ | $[1, 2, 4, 8]$ | $[1, 2, 4, 8]$ |
| | Condition-Only Plane Resolution | 25 | 25 | 25 |
| | Plane Embedding Dim | 32 | 32 | 32 |
| | Decoder MLP Hidden Dim | 128 | 128 | 128 |
| | Decoder MLP Layers | 3 layers | 3 layers | 3 layers |
| | **Total Parameters** | **12.1M** | **13.5M** | **6.8M** |
| FA-INR | Spatial Gate - Grid Resolution | 16 | 16 | 16 |
| | Spatial Gate - Grid Emb. Dim | 16 | 16 | 16 |
| | Condition - Feature Line Resolution | 16 | 10 | 10 |
| | Condition - Line Emb. Dim | 64 | 32 | 32 |
| | Num Experts | 8 | 5 | 5 |
| | Expert - Encoder MLP Layers | 4 layers | 2 layers | 2 layers |
| | Expert - Encoder MLP Dims | 512 | 256 | 256 |
| | Expert - Num KV Embeddings | 1024 | 512 | 512 |
| | Expert - KV Embeddings Dim | 32 | 16 | 16 |
| | Expert - KV Top K | 16 | 8 | 8 |
| | Decoder MLP Hidden Dim | 128 | 128 | 128 |
| | Decoder MLP Layers | 3 layers | 3 layers | 3 layers |
| | **Total Parameters** | **9.5M** | **1.0M** | **1.0M** |
| Explorable-INR | Spatial - Feature Grid Resolution | 64 | 64 | 48 |
| | Spatial - Grid Embedding Dim $d_{fs}$ | 32 | 32 | 32 |
| | Spatial - Feature Plane Resolution | 256 | 256 | 96 |
| | Spatial - Plane Embedding Dim | 32 | 32 | 32 |
| | Condition - Feature Line Resolution | 16 | 16 | 16 |
| | Condition - Line Embedding Dim | 16 | 16 | 16 |
| | Decoder MLP Hidden Dim | 128 | 128 | 128 |
| | Decoder MLP Layers | 3 layers | 3 layers | 3 layers |
| | **Total Parameters** | **14.7M** | **14.7M** | **4.5M** |
| DRR-Net (Ours) | Spatial - Feature Grid Multi-Res | $[32, 80, 112, 128]$ | $[64, 128]$ | $[16, 24, 32, 48]$ |
| | Spatial - Grid Embedding Dim | 2 | 4 | 2 |
| | Spatial $\pi$ - SSR Max Grid Res | - | 160 | 128 |
| | Spatial $\pi$ - P.E. Num Freqs $K_{pe}$ | 8 | 10 | 6 |
| | Spatial Refiner - Hidden Multiplier | 4 | 3.2 | 4 |
| | Spatial Refiner - Layers | 3 layers | 4 layers | 4 layers |
| | Condition - Feature Line Multi-Res | $[2, 4, 8, 16]$ | $[4, 8, 16, 32]$ | $[2, 4, 8, 16]$ |
| | Condition - Line Embedding Dim $d_{fc}$ | 4 | 4 | 2 |
| | Condition Refiner - Hidden Multiplier | 5.34 | 2 | 2.67 |
| | Condition Refiner - Layers | 4 layers | 3 layers | 2 layers |
| | Decoder MLP Hidden Dim | 128 | 128 | 128 |
| | Decoder MLP Layers | 3 layers | 2 layers | 3 layers |
| | **Total Parameters** | **8.9M** | **4.6M** | **0.9M** |

**CloverLeaf3D (Mallinson et al., 2013)** is a Lagrangian-Eulerian hydrodynamics simulation. The ensemble varies six parameters governing the initial state: three density values (*Density 1–3*) and three energy values (*Energy 1–3*). The parameter sampling is subject to a hierarchical constraint: the values are ordered such that $Energy_1 < Energy_2 < Energy_3$ and $Density_1 < Density_2 < Density_3$. Furthermore, a cross-variable constraint enforces that $Density_i < Energy_i$ for each corresponding component $i \in \{1, 2, 3\}$. The dataset comprises 600 ensemble members, with each simulation evolved for 200 timesteps.

## B.5    MODEL IMPLEMENTATION AND TRAINING PROCEDURES

**Model Hyperparameters.** For all baselines, we utilize their official implementations. We adhere strictly to the recommended hyperparameter settings provided in the original works for the evaluated datasets, such as Explorable-INR and FA-INR (Chen et al., 2025; Li et al., 2025). As K-Planes was not originally benchmarked on these specific datasets, we adapted its configuration to ensure its feature plane resolutions match the data resolution. We enumerate the detailed model configurations

Table 6: Training hyperparameters for each dataset.

| Dataset | epochs | condition batch size $N_c$ | coordinate batch size $N_x$ | condition-impsmp | coordinate-impsmp |
|---|---|---|---|---|---|
| Nyx | 50 | 16 | $2^{16}$ | ✓ | ✓ |
| MPAS-Ocean | 50 | 16 | $2^{16}$ | ✗ | ✓ |
| Cloverleaf3D | 50 | 32 | $2^{12}$ | ✗ | ✓ |

in Tab. 5. Full reproducibility details are also provided in the open-sourced experiment configuration files within the *configs* directory. For deeper architectural specifics of the baselines, we refer readers to the original publications (Chen et al., 2025; Li et al., 2025).

Regarding the DRR-Net specifications in Tab. 5, the *Multi-Res* entries denote the resolution hierarchy of the base structures. For instance, $[64, 128]$ indicates two distinct 3D feature grids with resolutions of $64^3$ and $128^3$, respectively. The input dimension to the refiner is determined by the concatenation of these embeddings following the $\pi$-transformations described in Sec. 3.3. Taking the Cloverleaf3D spatial encoder configuration as a concrete example, with 4 grids each of embedding size 2 and a P.E. frequency of 6, the final unified embedding size yields $4 \times 2 \times (2 \times 6) = 96$. As a GLU-based architecture, the refiner projects these features to a hidden width defined by the listed multiplier. In this case, a multiplier of approximately 2.67 expands the 96-dimensional input to a hidden bottleneck dimension of 256. We select these non-integer multipliers specifically to align the hidden dimensions with powers of 2 for computational efficiency.

**Training Procedure and Hyperparameters.** To ensure a fair comparison, all models, including our DRR-Net, are trained with the Adam optimizer (Kingma & Ba, 2014) using a base learning rate of $1 \times 10^{-4}$. All models use a cosine annealing scheduler (Loshchilov & Hutter, 2016) to decay the learning rate by two orders of magnitude over the course of training. Following the official implementation, the MLP-based spatial feature encoders in FA-INR use a separate, smaller learning rate of $1 \times 10^{-5}$ for the Nyx dataset. Each training batch consists of coordinates sampled from $N_c$ fields, with $N_x$ coordinates per field. Each INR surrogate model is trained to minimize the L2 loss for the ground truth scalar value of a scientific variable for the given coordinate and simulation parameter input. We present a comprehensive summary of all hyperparameters used for training every model on all datasets in Tab. 6.

For the detailed discussion on the hyperparameter settings for our data augmentation methods VP-S and VP-SC, we refer to the subsequent ablation study section in Sec. H.

**Data Preprocessing and Normalization.** Prior to training, we normalize all data inputs and outputs. Input spatial coordinates ($x$) are scaled to the range $[-1, 1]^{d_x}$ to be compatible with standard PyTorch grid sampling functions. Condition parameters ($c$) are min-max normalized to $[0, 1]^{d_c}$; the normalization range is determined by the global minimum and maximum values across both the training and testing sets to define the full range of interest for parameter space exploration. Finally, the ground truth field values ($v$) are also min-max normalized, where the statistics (min and max) are computed solely from the training set data following standard practice. We note one dataset-specific step: for the Nyx dataset, a logarithmic transform is applied to the Dark Matter Density values to better handle its large dynamic range.

**Importance-Driven Data Sampling.** For our main experiments, we follow the importance-driven sampling strategy of prior work (Chen et al., 2025). As reasoned by Chen et al. (2025), the technique could improve training by focusing on more complex or less frequent regions of the data. The importance score is based on the inverse frequency of values (Biswas et al., 2018; 2023). For each ensemble member, a histogram of its field values is constructed, and each coordinate's importance is defined by the inverse frequency of its value. The importance of an entire member field is the sum of its coordinate scores, normalized across all members. In order to keep experiment settings consistent with Explorable-INR and FA-INR, for the conditional generalization task, we also use these scores to preferentially sample both the ensemble members and the coordinates within them.

**Uniform Random Sampling for the Super-Resolution Task.** For the novel spatio-conditional generalization task presented in Sec. 4.3, we do not apply importance sampling. As this is a new

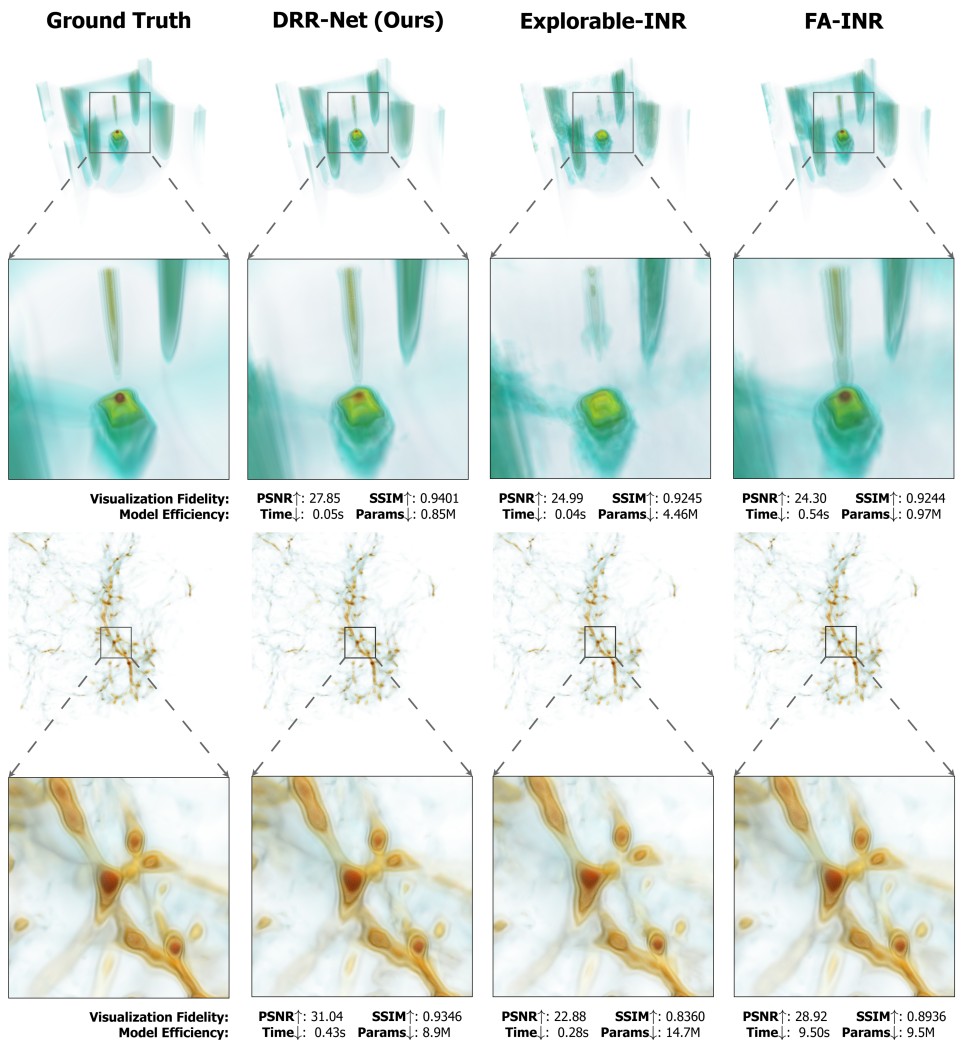

Figure 4: Visual comparison on an unseen field from the Cloverleaf3D (top) and Nyx (bottom) dataset. Reconstructed $128^3$ points in just 0.05 seconds, our DRR-Net better captures complex high-energy and high-density structures in respective datasets than competing state-of-the-art surrogates, demonstrating a superior balance of fidelity and speed.

evaluation task, we opt for a simpler, strong baseline using uniform random sampling across both the conditional and spatial domains.

## C  DATA VISUALIZATIONS FOR QUALITATIVE EVALUATION

This section provides additional qualitative comparisons for the conditional generalization task discussed in Sec. 4.2. We present volume renderings (Max, 2002) of unseen ensemble members from both the Cloverleaf3D and Nyx datasets. For the unstructured mesh dataset MPAS-Ocean, we visualize a depth slice of the temperature field over the globe.

### C.1  CLOVERLEAF3D VISUALIZATION

As shown in Fig. 4 (top), the visualization for the Cloverleaf3D dataset confirms DRR-Net's superior fidelity. Our method accurately captures the complex, high-energy structures of the simulation. In contrast, these fine details are poorly resolved by competing methods, highlighting DRR-Net's effectiveness in representing high-frequency features.

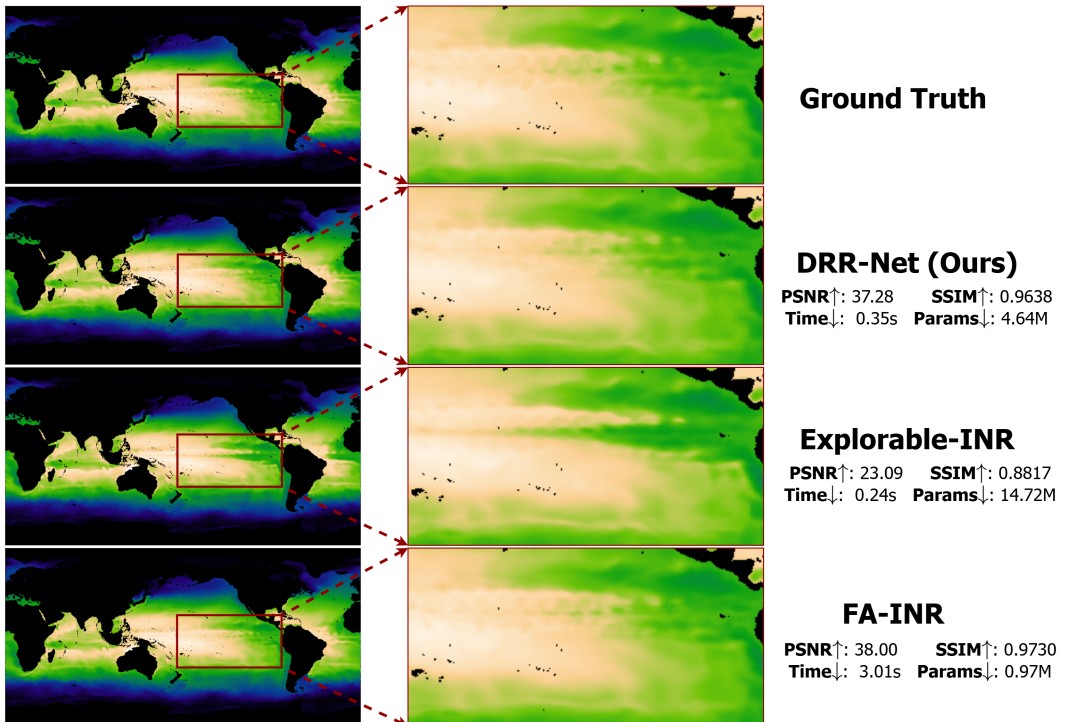

Figure 5: Qualitative comparison on the unstructured MPAS-Ocean dataset with one depth layer extracted. The visualization highlights the effectiveness of the DRR paradigm: while the standard embedding-based Explorable-INR suffers from severe artifacts, our DRR-Net successfully captures fine temperature variations, demonstrating the framework's power to enhance a grid-based backbone on challenging, non-Cartesian data, while maintaining efficiency.

## C.2 NYX VISUALIZATION

Fig. 4 (bottom) presents a qualitative comparison for the Nyx dataset, focusing on a high-density region of interest containing halo features. Both DRR-Net and the high-fidelity baseline FA-INR provide the most faithful reconstructions of the complex filamentary structures. However, DRR-Net's result can be perceptually closer to the ground truth to a minor extent, an observation corroborated by its superior PSNR and SSIM scores.

Beyond visual fidelity, this comparison underscores DRR-Net's significant efficiency advantage. Reconstructing the full $256^3$ field takes 0.43 seconds, making it over 22× faster than FA-INR. This level of performance is critical for practical scientific applications, enabling interactive data exploration. Furthermore, this speed is achieved without any hardware-specific optimizations. We project that by incorporating low-level acceleration techniques, such as the customized CUDA kernels used in modern INRs (Müller et al., 2022), the DRR paradigm could support real-time analysis of ensemble simulations. This would empower domain scientists with truly interactive workflows, such as iterative volume visualization and notebook-style exploratory analysis, for rapid hypothesis testing.

## C.3 MPAS-OCEAN VISUALIZATION

To understand the fidelity of reconstructed data for MPAS-Ocean, we extract a single layer from the depth or radius dimension of the mesh. This layer contains the highest-frequency features, characterized by the greatest temperature variations across the sphere, from one of the ensemble members. Presented in Fig. 5, we observe that both DRR-Net and the MLP-based FA-INR produce high-fidelity reconstructions that are visually close to the ground truth.

The more insightful comparison, however, is between DRR-Net and the standard embedding-based Explorable-INR. Both models share a similar grid-based backbone, which is inherently challenged

by the unstructured mesh of the MPAS-Ocean simulation. Despite this shared disadvantage, DRR-Net successfully captures the fine-grained and wavy temperature variations in green, particularly in the zoomed-in view. In contrast, Explorable-INR fails to resolve these details, resulting in a blurry reconstruction. This significant difference in quality demonstrates that the DRR framework is highly effective at enhancing the representational power of the underlying embedding structure, enabling it to overcome the challenges of non-Cartesian data. Furthermore, while DRR-Net's visual fidelity is comparable to FA-INR, it achieves this result over $8\times$ faster, showcasing a superior balance between reconstruction quality and inference speed.

## D    CONDITIONAL GENERALIZATION ON LOW-RESOLUTION FIELDS

Table 7: Conditional generalization results for models trained on data with $\times 2$ reduced spatial resolution. Consistent with findings in Sec. 4.3, DRR-Net achieves the highest fidelity and efficiency. Notably, DRR-Net exhibits a significant expressiveness advantage in modeling the Nyx dataset in the low-resolution setup.

| Dataset | Model | Unseen Fields (Rel L2↓ / PSNR↑ / SSIM↑) | Trained Fields (Rel L2↓ / PSNR↑ / SSIM↑) | Inference Time (sec) | # Params |
|---|---|---|---|---|---|
| Nyx | K-Planes | 1.49e-01 / 29.44 / 0.667 | 1.46e-01 / 29.91 / 0.688 | 11.4s | **3.9M** |
| | FA-INR | 2.78e-02 / 44.00 / 0.987 | 2.76e-02 / 44.38 / 0.988 | 156.3s | 9.5M |
| | Explorable-INR | 4.46e-02 / 39.89 / 0.979 | 3.29e-02 / 42.86 / 0.984 | **6.5s** | 10.0M |
| | DRR-Net (ours) | **1.11e-02 / 51.95 / 0.998** | **1.15e-02 / 51.99 / 0.998** | 8.0s | 8.7M |
| Cloverleaf3D | K-Planes | 4.33e-01 / 35.32 / 0.903 | 4.15e-01 / 35.94 / 0.909 | 16.1s | 3.1M |
| | FA-INR | 9.92e-02 / 48.12 / 0.991 | 6.24e-02 / 52.39 / 0.995 | 53.8s | 1.0M |
| | Explorable-INR | 1.17e-01 / 46.65 / 0.989 | 7.48e-02 / 50.82 / 0.993 | 6.2s | 4.0M |
| | DRR-Net (ours) | **9.38e-02 / 48.60 / 0.991** | **5.96e-02 / 52.80 / 0.995** | **6.1s** | **0.9M** |

While the spatio-conditional generalization task in Sec. 4.3 focused on super-resolution, which involved training on $\times 2$ downsampled data and evaluating on unseen full-resolution fields, here we evaluate the models directly in the low-resolution domain. Since the reduced resolution can simplify or even eliminate fine-grained high-frequency features, this theoretically presents a simpler learning objective. Consequently, evaluating the surrogate model in this low-resolution setup serves as a valuable case study to probe the model's capability to accurately capture simpler simulation scenarios.

The quantitative results, presented in Tab. 7, demonstrate that DRR-Net achieves state-of-the-art fidelity across all metrics. On the Nyx dataset, our method significantly outperforms the nearest competitor, FA-INR, reducing the relative $L_2$ error on unseen fields by approximately $60\%$ (1.11e-02 vs. 2.78e-02) and providing a substantial boost in PSNR from $44.00$ dB to $51.95$ dB. Notably, DRR-Net's accuracy in modeling the $128^3$ low-resolution Nyx data is significantly higher than in the high-resolution scenario ($\approx 42$ dB), confirming the substantial representational capacity afforded by the proposed DRR architectural paradigm, the combination of refiner networks and embedding structures. A similar trend is observed on Cloverleaf3D, where DRR-Net consistently attains the highest fidelity for both trained and unseen fields. Crucially, this performance is achieved with high efficiency; DRR-Net maintains inference latency comparable to the lightweight Explorable-INR ($\approx$ 6–8 seconds) and is significantly faster than FA-INR, offering inference speedups of approximately $9\times$ on Cloverleaf3D and $19\times$ on Nyx.

## E    ABLATION STUDY: EFFECT OF EMBEDDING REFINEMENT

To isolate the effect of the non-linear refinement from the DRR formulation, we conduct an ablation study, starting with an Embedding-Only baseline and progressively adding our refiner modules, including the parameter-free $\pi$ functions and the refiner network. The results, presented in Tab. 8, demonstrate the contribution of each component.

**Spatial Refiner (SR).** The SR provides the most significant and consistent performance gain. Across all three datasets, incorporating the spatial refiner boosts the PSNR of the embedding-only baseline by a large margin, ranging from +2.1 dB on Nyx to +3.2 dB on MPAS-Ocean. This result pro-

Table 8: Ablation study on the DRR components, demonstrating that the embedding refinement procedure provides considerable performance gains over the embedding-only baseline.

| Dataset | DRR-Net Variants | Rel L2↓ | PSNR↑ | SSIM↑ | TFLOPs / $10^9$ pts ↓ | # Params |
|---------|------------------|---------|-------|-------|----------------------|----------|
| Nyx | Embedding-Only | 4.02e-02 | 42.64 | 0.979 | 40.1 | 8.1M |
| | + Spatial Refiner | **3.17e-02** | **44.71** | **0.986** | 57.2 | 8.7M |
| | + Condition Refiner | 3.18e-02 | 44.69 | 0.986 | 57.2 | 8.9M |
| MPAS-Ocean | Embedding-Only | 1.24e-02 | 45.22 | - | 25.8 | 3.57M |
| | + Spatial Refiner | 8.54e-03 | 48.43 | - | 47.3 | 4.57M |
| | + Condition Refiner | **7.76e-03** | **49.26** | - | 47.3 | 4.64M |
| Cloverleaf3D | Embedding-Only | 1.43e-01 | 45.44 | 0.987 | 40.1 | 0.4M |
| | + Spatial Refiner | 1.05e-01 | 48.11 | 0.992 | 52.6 | 0.8M |
| | + Condition Refiner | **9.81e-02** | **48.69** | **0.994** | 52.6 | 0.9M |

vides clear evidence for the fundamental effectiveness of the DRR paradigm in promoting a more expressive spatial representation than can be achieved with embeddings alone.

**Condition Refiner (CR).** The addition of the CR constitutes the full DRR-Net and further improves performance on the MPAS-Ocean and Cloverleaf3D datasets by +0.83 dB and +0.58 dB, respectively. This demonstrates the value of explicitly refining the conditional features for these more complex ensemble problems. Interestingly, on the Nyx dataset, the CR does not yield an additional gain. We hypothesize this is due to the relatively simple conditional space of this dataset, where the multi-scale base embeddings are already sufficient to capture the inter-parameter variations.

Overall, this study confirms that both spatial and condition embedding refinement components are crucial for achieving state-of-the-art performance. The MPAS-Ocean results are particularly telling: the massive performance jump over the Embedding-Only baseline demonstrates that our refiners are highly effective at compensating for the structural mismatch between a Cartesian base grid and an unstructured dataset, significantly enhancing the model's representational power.

# F    ABLATION STUDY: NON-PARAMETRIC REFINEMENT WITH $\pi$ FUNCTIONS

The parameter-free preprocessing function, $\pi$, is a pivotal component of the DRR framework, designed to generate a higher-capacity input representation without increasing the model's parameter count. In this section, we analyze the efficacy of $\pi$-transformations (P.E. and Structural Super-Resolution) both as an isolated non-parametric refinement strategy and as a pre-conditioner for the refiner network.

We categorize the results into two distinct regimes: (1) The Low-Capacity Regime, involving the original low-resolution base embeddings; and (2) The High-Capacity Regime, involving the $\pi$-transformed embeddings. This comparative analysis in Sec. F.1 sheds light on the critical role of the non-parametric $\pi$ functions: they not only enhance the embedding structure on their own as a powerful non-parametric realization of DRR but also serve as a necessary utility to enable the learning capability of the learnable refiner network. Finally, in (3) $\pi$ Hyperparameter Studies on Full DRR-Net, we vary the hyperparameters of $\pi$ to analyze their impact on fidelity in Sec. F.2 and Sec. F.3.

## F.1    EFFECT OF $\pi$ FUNCTIONS FOR DRR

Tab. 9 quantifies the impact of the $\pi$ preprocessing function by comparing model performance in its absence versus its presence. This comparison highlights two critical roles of the $\pi$ function: serving as a performant refinement function and ensuring deep refinement as well as generalization with the refiner.

**(1) The Necessity of $\pi$ for Generalization.** In the absence of $\pi$ ("No Refinement"), the refiner is forced to operate on the base embeddings in their native state. This presents two critical bottlenecks: first, the base structure implies a highly constrained feature space (e.g., the spatial branch yields a concatenated embedding vector of only dimension 8); second, the coarse resolution of the original grids provides a sparse set of unique feature vectors, resulting in inadequate learning signals for

Table 9: Ablation study analyzing the impact of the refiner on low-capacity base embeddings versus $\pi$-transformed embeddings. Accuracy metrics are reported for the unseen simulation conditions. We categorize the results into two groups, using the base embedding and the $\pi$-transformed embedding as their respective baselines. The results demonstrate that the proposed $\pi$-transformation not only enhances embedding capacity on its own but also acts as a pivotal enabler, unlocking the full potential of the refiner network.

| Dataset | Refinement Regime | Rel L2↓ | PSNR↑ | SSIM↑ | TFLOPs / $10^9$ pts ↓ | # Params |
|---|---|---|---|---|---|---|
| Nyx | No Refinement | **4.02e-02** | **42.64** | **0.979** | 40.1 | 8.1M |
| | + Refiners | 4.48e-02 | 41.70 | 0.975 | 40.1 | 8.3M |
| | $\pi$ Refinement | 4.04e-02 | 42.60 | 0.978 | 55.9 | 8.1M |
| | + Refiners | **3.18e-02** | **44.69** | **0.986** | 57.2 | 8.9M |
| MPAS-Ocean | No Refinement | **1.24e-02** | **45.22** | - | 25.8 | 3.57M |
| | + Refiners | 1.38e-02 | 44.29 | - | 25.8 | 3.65M |
| | $\pi$ Refinement | 1.01e-02 | 47.00 | - | 45.8 | 3.57M |
| | + Refiners | **7.76e-03** | **49.26** | - | 47.3 | 4.64M |
| Cloverleaf3D | No Refinement | **1.43e-01** | **45.44** | **0.987** | 40.1 | 0.37M |
| | + Refiners | 1.62e-01 | 44.32 | 0.985 | 40.1 | 0.41M |
| | $\pi$ Refinement | 1.13e-01 | 47.48 | 0.991 | 51.7 | 0.37M |
| | + Refiners | **9.81e-02** | **48.69** | **0.994** | 52.6 | 0.82M |

optimizing a deep network. Consequently, we observe that adding a refiner in this regime degrades performance on unseen members; for instance, the PSNR on the Nyx dataset drops from 42.64 dB to 41.70 dB. Ultimately, the refiner is starved of sufficient data points and representational capacity in this base regime, causing it to overfit to the constrained features rather than learning generalizable data patterns.

**(2) $\pi$ as an Enabler for Refinement.** Introducing the $\pi$ function fundamentally alters the learning dynamic, yielding two distinct benefits.

First, we observe the efficacy of non-parametric transformations within the DRR paradigm. Comparing the "$\pi$ Refinement" rows to the "No Refinement" baselines in Tab. 9 reveals that the $\pi$-transformation alone consistently improves fidelity. This demonstrates that the structural priors introduced by $\pi$, specifically the spatial smoothness induced by super-resolution and the non-linear multi-scale frequency expansion provided by P.E., yield immediate representational benefits. This establishes the inherent value of our core contribution, the DRR paradigm: even in the absence of additional learnable parameters, decoupling the inference structure via deterministic refinement produces a significantly more expressive representation than the base embedding structure alone.

Second, and more critically, we identify a synergistic enablement of the learnable refiner. By projecting the sparse base embeddings into a high-dimensional, high-resolution manifold, $\pi$ provides the rich abundance of unique feature vectors necessary to optimize a deep neural network. In this expanded regime, the refiner is no longer bottlenecked by data scarcity. Instead, it successfully exploits the increased capacity to deliver significant additive gains over the $\pi$-only baseline ("$\pi$ Refinement + Refiners"). For example, on the MPAS-Ocean dataset, the refiner reduces the relative L2 error from 1.01e-02 ($\pi$-only) to 7.76e-03 (Full DRR-Net). Thus, the results reveal that $\pi$ is not merely a preprocessing step, but the foundational enabler that allows the refiner to transition from overfitting to learning generalized, state-of-the-art representations.

For a qualitative comparison illustrating how the learnable refiner overcomes the specific limitations of the $\pi$-only regime, such as high-frequency noise and aliasing, please refer to the visual analysis in Sec. G.

## F.2 FEATURE UPSAMPLING HYPERPARAMETER: P.E. FREQUENCIES

We study the effect of the level of frequencies used in P.E. proposed by Mildenhall et al. (2020) for feature upsampling, with results shown in Fig. 6. The charts reveal a consistent pattern across all datasets: performance steadily increases as the number of frequencies rises to a moderate level, around 6 to 8, after which it either plateaus or degrades sharply.

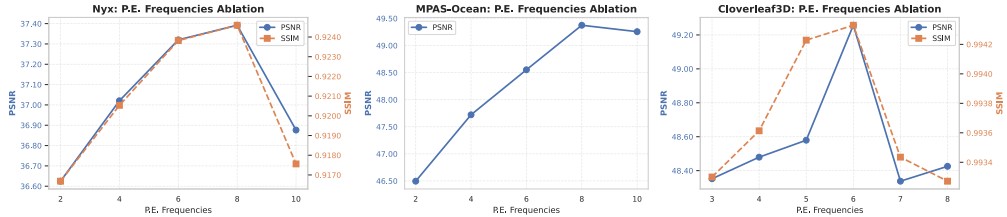

Figure 6: Ablation studies of the frequency hyperparameter of Positional Encoding for feature upsampling. A moderate frequency level around 6 is observed to consistently improve the efficacy of DRR-Net.

This trend supports a hypothesis that the number of P.E. frequencies controls a critical trade-off. Initially, increasing the frequency upsamples the features into a higher-dimensional space. This provides the refiner with a richer representation, giving it more capacity to capture the complex, multi-frequency patterns present in the data.

However, this benefit becomes counterproductive beyond an optimal point for two primary reasons. First, an excessively high-frequency feature space may introduce a signal bias that does not match the intrinsic properties of the data with predominantly lower frequency patterns, leading to overfitting or aliasing, which is also reported by Zhang et al. (2024). Second, the feature dimensionality can become too high to be optimized effectively given a finite amount of training data, creating a more difficult learning problem.

Summarizing from the results, a low to moderate frequency level from 2 to 6 for the P.E. is expected to be a safe choice and effectively lift the embeddings to a richer multi-frequency representation for the refiner to operate on. On the other hand, an aggressive frequency level can potentially decrease the performance, which is only advisable for datasets known to contain very high-frequency features.

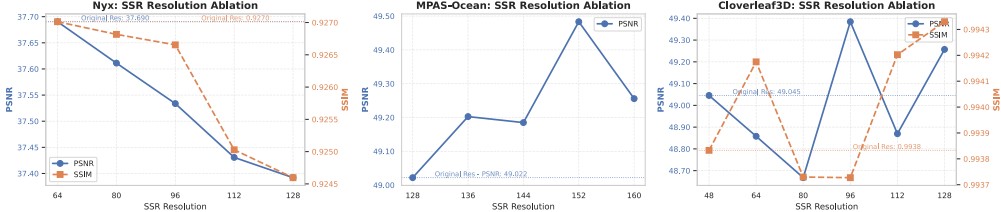

Figure 7: Comparisons of DRR-Net with different maximum resolution for the structural superresolution (SSR). The baseline configuration without SSR is presented. SSR exhibits the most advantage for MPAS-Ocean and Cloverleaf3D.

### F.3 STRUCTURAL SUPER-RESOLUTION HYPERPARAMETER: MAXIMUM RESOLUTIONS

We conduct an ablation study to investigate the impact of the maximum resolution used in Structural Super-Resolution (SSR), with results presented in Fig. 7. Our findings indicate that the effectiveness of the parameter-free SSR can be dataset-dependent, offering significant benefits in some cases while being less effective in others.

For the MPAS-Ocean and Cloverleaf3D datasets, applying SSR yields clear performance gains over the baseline configuration. As shown in the charts, an appropriately tuned maximum resolution, such as 152 for MPAS-Ocean and 128 for Cloverleaf3D, leads to a notable improvement in both PSNR and SSIM over the baseline without SSR. In contrast, for the Nyx dataset, tasking the refiner with learning from an interpolated high-resolution grid can slightly degrade quality compared to learning from the original grids directly.

This divergence appears to stem from the interaction between interpolation and the signal's intrinsic properties. For datasets like MPAS-Ocean and Cloverleaf3D, the underlying signal is relatively coarse and spatially smooth outside of small regions of interest. In these cases, an interpolated grid

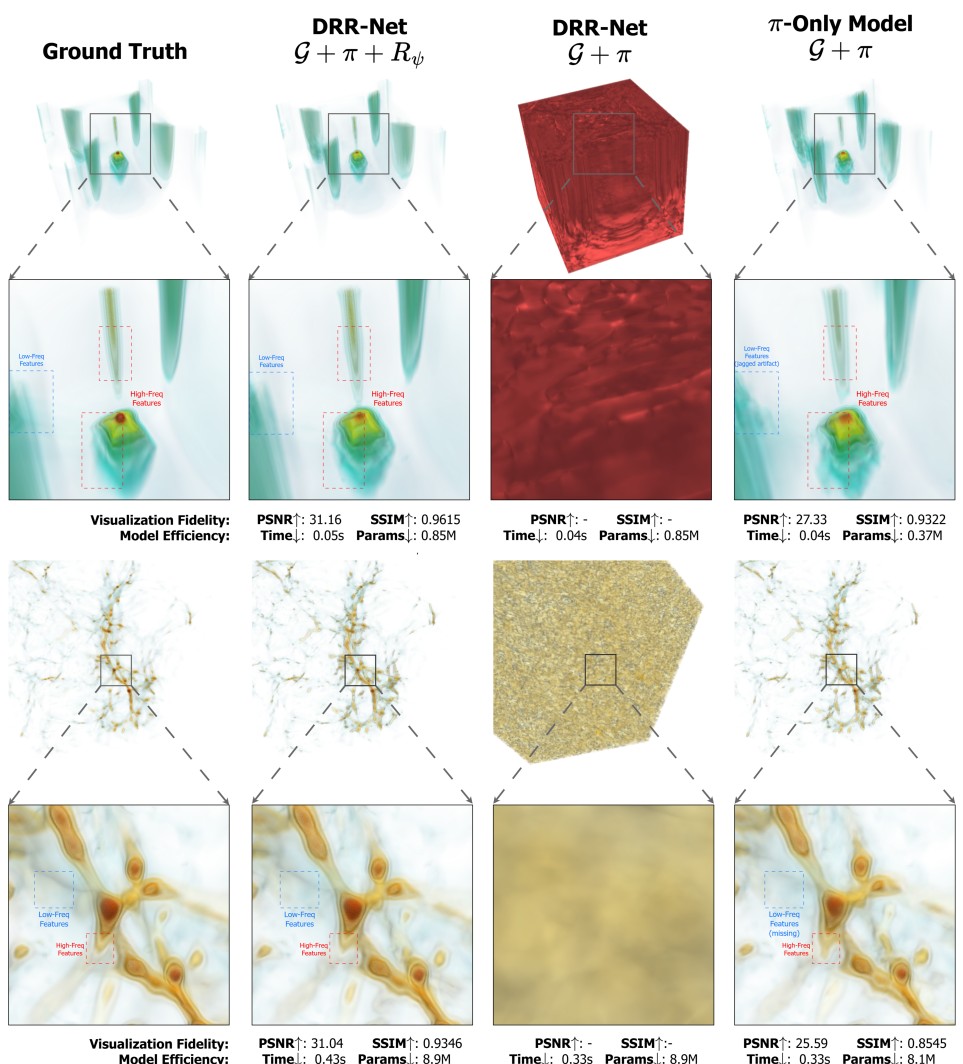

Figure 8: Visual ablation on unseen Cloverleaf3D (top) and Nyx (bottom) members with volume rendering for understanding the role of refiner in the proposed DRR paradigm. Columns: (1) Ground Truth; (2) Full DRR-Net; (3) Inference with refiner disabled; (4) $\pi$-Refinement baseline. Two key observations: (1) Disabling the refiner yields severe artifacts (Col. 3), confirming it is an integral part of the learned representation in DRR-Net. (2) Compared to the model trained only with the $\pi$ refinement (Col. 4), the Full DRR-Net (Col. 2) effectively suppresses noisy artifacts in smooth regions while sharply resolving high-frequency details in shades of red, validating its superior multi-scale feature fusion.

provides a reasonable approximation of the ground truth. The refiner's task is then a manageable problem of adding high-frequency details to a strong baseline.

The Nyx data presents a different scenario. Although Fig. 4 only renders the high-density regions, the full data field is dense with complex, high-frequency patterns even for the regions with lower dark matter density values. A coarse grid of such a signal is a poor representation that loses significant information. The subsequent upsampling by interpolation acts as a low-pass filter and creates a smooth field that is an inaccurate approximation of the true signal. This presents the refiner with a difficult, ill-posed learning problem of attempting to reconstruct a vast amount of lost high-frequency detail. This task can be less effective than learning directly from the original multi-resolution grids that do not suffer from this interpolation-induced information loss.

## G    UNDERSTANDING THE EFFECT OF REFINER WITH VISUALIZATION

To complement our quantitative ablation studies on the proposed DRR components, including the complete refinement components (Sec. E) and non-parametric $\pi$ refinement functions (Sec. F.1), we present a visual analysis in Fig. 8 to provide a holistic understanding of the refiner network's role in enhancing representation quality. We visualize unseen members from the Cloverleaf3D (top row) and Nyx (bottom row) datasets with the volume rendering algorithm (Max, 2002) across four configurations: (1) Ground Truth; (2) the Full DRR-Net; (3) the Full DRR-Net with the refiner disabled at inference time; and (4) a baseline trained with $\pi$-Refinement only. This comparison yields two prominent observations regarding the nature and quality of the learned representation.

**Visual Validation of Synergistic Division of Labor.** The third column displays the inference result of a fully trained DRR-Net when the refiner module is bypassed. The resulting images consist almost entirely of unintelligible artifacts. This result visually confirms the *synergistic division of labor* hypothesis proposed in Sec. 3.2.2. It demonstrates that the base embedding $\mathcal{G}$ and the refiner $R_\psi$ become tightly coupled during optimization: the base embeddings have specialized to encode a latent pre-conditioner, likely capturing high-frequency local properties, that relies on the refiner to resolve global consistency and feature fusion. Consequently, the refiner is not a redundant post-processing step, but an integral component required to decode this specialized base representation into a valid physical field.

**Improved Multi-Scale Feature Fusion.** A direct comparison between the Full DRR-Net (second column) and the $\pi$-Only baseline (fourth column) highlights the refiner's superior capability in multi-scale feature fusion. The $\pi$-Only model, relying on deterministic upsampling, exhibits two characteristic failure modes: it generates spurious high-frequency noise, manifesting as jagged grid artifacts, in spatially smooth regions, such as the peripheral areas of the Cloverleaf3D simulation. Simultaneously, it produces over-blurry reconstructions in regions containing genuine high-frequency details, such as the central high-energy structures.

In contrast, the Full DRR-Net successfully acts as a selective fusion mechanism. It suppresses noisy representation in smooth areas while preserving and sharpening true high-frequency signals, as observed in the thin high-energy structures from the zoomed-in view of Cloverleaf3D. This validates the refiner's capacity to distinguish between aliasing artifacts and physical features, resulting in a reconstruction that is both smoother in homogeneous regions and sharper at high-frequency structures.

## H    ABLATION STUDY: DATA AUGMENTATION HYPERPARAMETERS

The Variational Pairs (VP) data augmentation strategy, introduced in Sec. 3.4, is a general tool for improving the generalization of INR surrogates from the perspective of the data. Its effectiveness can be dependent on the tuning of its noise generation hyperparameters. This section provides a detailed ablation study of these parameters to offer practical guidance for their use. Our study focuses on the noise truncation threshold, $\tau$, which governs the aggressiveness of the VP augmentation. The standard deviation of the noise, $\sigma$, is kept proportional to this threshold, making $\tau$ the primary hyperparameter controlling the trade-off between data diversity and sample reliability. All studies are conducted using our DRR-Net on the Nyx dataset.

We analyze the impact of the threshold on three augmentation variants:

- **VP-S:** The baseline spatial-only augmentation, where the spatial threshold is kept at a fixed, conservative value based on the grid resolution.
- **VP-C:** A conditional-only variant designed to isolate the effect of condition noise. This variant was not presented in Sec. 3.4 and is used here for analytical purposes.
- **VP-SC:** The full spatio-conditional augmentation.

### H.1    VP-S: COORDINATE NOISE THRESHOLD

The ablation study for the spatial-only augmentation, VP-S, is presented in Fig. 9 (left). The tested coordinate noise thresholds are derived from the vertex spacing of hypothetical grid resolutions

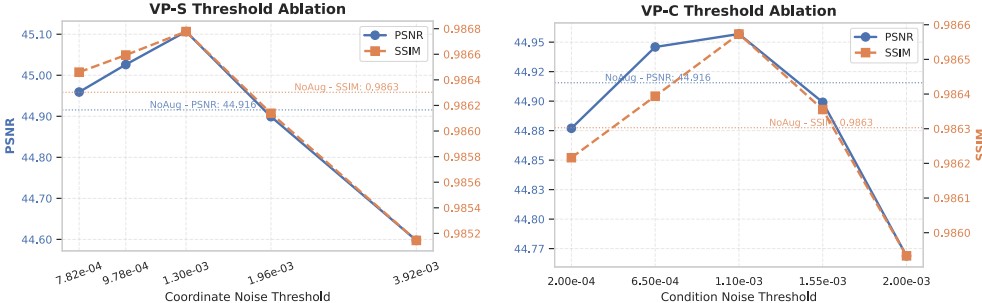

Figure 9: Comparison of the noise threshold for the isolated spatial-only and condition-only VP methods in the Nyx dataset. We observe that incorporating VP-S or VP-C can both improve the accuracy over a non-augmented baseline.

finer than the native $256^3$ grid with threshold 3.92e-03. A smaller threshold value on the chart, such as 1.30e-03, corresponds to a finer hypothetical grid and thus represents a more conservative and constrained noise range. We observe a clear trend where performance peaks at this conservative threshold and declines as the noise range becomes either too small or too large.

An optimally chosen threshold provides a performance improvement of approximately 0.2 dB in PSNR over the non-augmented baseline. This result shows that a conservative and constrained noise range can effectively improve DRR-Net for the Nyx dataset. Since VP generates new values via interpolation, limiting the perturbation range is critical. A constrained range ensures the augmented samples remain in a region where the simple interpolation function remains a reliable approximation of the true data patterns.

Conversely, the performance degradation at higher noise thresholds indicates that allowing augmented samples to span broader spatial regions can exacerbate the distribution mismatch between the interpolated values and the ground truth. This introduces unreliable training signals and harms performance. Therefore, it is advisable to keep the spatial noise constrained to a relatively small range, such as one-quarter of the native grid spacing.

### H.2 VP-C: CONDITION NOISE THRESHOLD

Defining a meaningful noise threshold for conditional parameters is more nuanced than for regular spatial grids due to the irregular sampling of the parameter space. To establish a principled range of conservative thresholds, we first analyze the distances for each simulation parameter across all ensemble members individually. For each parameter, we sort all sample values and find the minimum distance between any two adjacent points. This gives us a set of minimum distances, one for each parameter dimension. We then define our experimental range by taking the minimum and maximum of these values and uniformly sampling five thresholds within this range. This procedure allows us to test a spectrum of conservative noise levels, from one based on the tightest sample packing in any dimension to one based on the loosest.

The results for the conditional-only augmentation, VP-C, are presented in Figure 9 (right). The trend is remarkably similar to that of VP-S, confirming that conditional augmentation can effectively boost model performance. Performance peaks at a moderate threshold of 1.10e-03 and is worse on either side of the magnitude range, with the lowest and highest thresholds performing below the non-augmented baseline.

These results imply the same principle observed in VP-S: a sufficiently constrained noise level can provide diverse learning signals beyond the original sparse parameter samples, while still ensuring the reliability of the conditional interpolation. However, since sampling strategies for conditional parameters can vary significantly between simulations and may not resemble a regular grid, devising a reasonable noise level can be more challenging and may benefit from a domain-specific understanding of the parameter space.

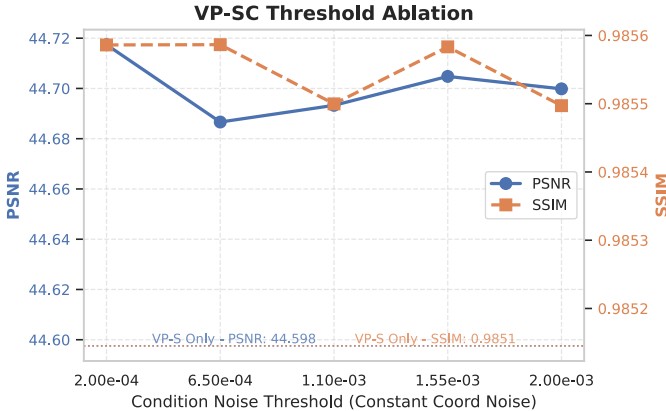

Figure 10: Ablation study on varying the condition noise thresholds for VP-SC with a constant spatial noise level. Results show that adding the condition noises can consistently improve DRR-Net quality marginally, compared to VP-S.

### H.3    VP-SC: Spatio-Conditional Noise Threshold

Finally, we analyze the full spatio-conditional augmentation, VP-SC. In this study, presented in Fig.10, we vary the conditional noise threshold using the same range as the VP-C experiment, while keeping the spatial noise at a constant threshold 3.92e-03. The spatial noise level is also the setup for the main evaluations in Sec 4.2 and Sec. 4.3.

The results show that adding conditional noise on top of spatial noise provides a consistent, though marginal, performance improvement over using only spatial augmentation. A key observation is that the model is much more tolerant to the choice of the conditional threshold in this combined setting. Unlike the isolated VP-C study, all tested threshold values in the VP-SC configuration outperform the VP-S baseline. This suggests that the presence of spatial noise may have a regularizing effect that makes the model more robust to variations in the conditional augmentation.

### H.4    Final Remarks on VP Augmentation

Our ablation studies on the Variational Pairs augmentation strategy reveal several key takeaways for practical application.

Spatial-only augmentation (VP-S) serves as a strong and reliable baseline. For simulations on regular grids, its noise threshold is straightforward to configure, requiring minimal hyperparameter tuning. Conditional-only augmentation (VP-C) can also be effective, but determining an optimal threshold is more challenging due to the irregular nature of parameter spaces and may benefit from domain-specific knowledge.

The full spatio-conditional variant (VP-SC) offers a path to the best augmentation result. It can provide an additive improvement over VP-S, a pattern observed across all models in Sec. 4.4. Furthermore, our study here shows that VP-SC is more tolerant to the specific choice of the conditional noise threshold. Therefore, we recommend VP-S as a simple and robust starting point, with VP-SC as a more advanced option to extract further performance, especially in scenarios where tuning conditional noise is complex.

## I    Application to Vision and Graphics Tasks

To demonstrate the versatility of the DRR paradigm, we extend its application beyond scientific simulations to common vision and graphics tasks. This section evaluates DRR-enhanced INRs for neural image representation (Sec. I.1) and Neural Radiance Fields (Sec. I.2).

Table 10: Performance comparison of various INRs with and without DRR for two Gigapixel images. Rows marked "+ DRR" show the change (Δ) relative to the base model above. Green indicates an improvement. Strong baseline models, including NGP and SIREN, are also evaluated to serve as a reference.

| Gigapixel Image | Model | PSNR↑ | SSIM↑ | LPIPS↓ | Inference Time (sec) | # Params |
|---|---|---|---|---|---|---|
| Girl with a Pearl Earring | Grid | 25.80 | 0.9180 | 0.2504 | 0.86s | 4.2M |
| | + DRR | +0.12 | +0.0016 | -0.0040 | 1.16s | 4.3M |
| | MRGrid | 25.86 | 0.9191 | 0.2468 | 1.17s | 4.2M |
| | + DRR | -0.02 | -0.0012 | +0.0055 | 2.28s | 4.3M |
| 23466×20000 | NGP | 26.45 | 0.9154 | 0.2235 | 0.90s | 5.2M |
| | SIREN | 24.64 | 0.8713 | 0.2827 | 27.82s | 2.1M |
| Tokyo | Grid | 27.13 | 0.9396 | 0.1700 | 2.13s | 4.2M |
| | + DRR | +1.11 | +0.0108 | -0.0164 | 2.58s | 4.3M |
| | MRGrid | 28.08 | 0.9493 | 0.1476 | 2.94s | 4.2M |
| | + DRR | +0.92 | +0.0070 | -0.0119 | 5.55s | 4.3M |
| 21450×56718 | NGP | 25.63 | 0.8978 | 0.1873 | 2.22s | 5.2M |
| | SIREN | 14.31 | 0.1061 | 0.7288 | 71.63s | 2.1M |

## I.1 GIGAPIXEL IMAGE REPRESENTATION

**Experimental Setup** We evaluate the DRR paradigm on the task of gigapixel image representation, where an INR is trained to map 2D pixel coordinates to their corresponding RGB values. We apply DRR to two embedding-based backbones: a single-resolution feature grid (Grid) and a multi-resolution grid (MRGrid). The models contain a feature grid encoder and a 2-layer lightweight MLP decoder for prediction. Both structural super-resolution and P.E. feature upsampling are applied prior to the refinement. These DRR-enhanced models are compared against their original counterparts. Although the experiment aims to study the effect of the DRR paradigm on different embedding-based models, we present two effective baselines, NGP (Müller et al., 2022) and SIREN (Sitzmann et al., 2020), to provide performance contexts. The evaluation is conducted on two images with different characteristics: the relatively smooth "Girl with a Pearl Earring" at 23466 × 20000 resolution and the more challenging "Tokyo" cityscape with significantly more objects and textures at the resolution of 21450 × 56718.

**Quantitative Evaluation.** The results in Tab. 10 show that DRR can substantially improve reconstruction quality, particularly for images with complex, high-frequency details. On the visually intricate "Tokyo" cityscape, applying DRR provides a significant performance boost to both base models. It improves the PSNR by +1.11 dB for the single-resolution Grid and +0.92 dB for the Multi-Resolution Grid (MRGrid). We later show that this creates perceptually significant improvement of the reconstruction in the visualizations. Notably, both DRR-enhanced models considerably outperform the strong NGP baseline on this challenging image.

The results on the "Girl with a Pearl Earring" image, which contains smoother content, are more nuanced. While DRR provides a modest improvement to the simple Grid model, it slightly degrades the performance of the more powerful MRGrid. This suggests the benefit of the DRR refiner is most pronounced when the base representation has room for improvement or when the signal is highly complex. For simpler signals where a strong base model like MRGrid may already be sufficient to capture the content, the additional capacity of the refiner might not provide a significant benefit.

**Qualitative Evaluation.** Fig.11 presents a qualitative comparison on the complex "Tokyo" image, showing cropped views at progressive levels of detail. The visualization studies the effect of DRR on the MRGrid model and includes the NGP model for reference. The results reveal a significant visual improvement offered by the DRR paradigm.

The first evidence of reconstruction loss is the level of image noise. NGP exhibits the most noise, creating jagged and inconsistent textures, followed by the base MRGrid. The improvement from DRR is evident even at moderate magnification from the level 2 images in the second column,

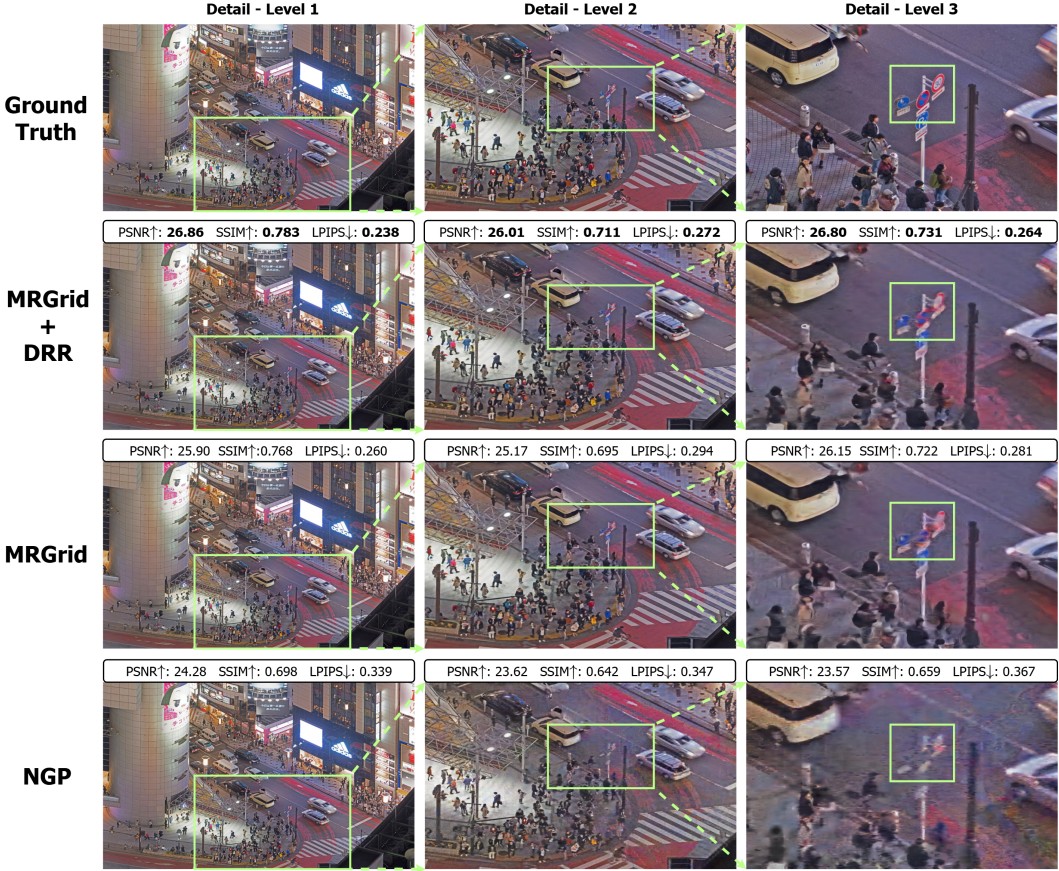

Figure 11: Visualization of a Gigapixel image with a highly complex object composition at a progressive level of detail. The addition of DRR significantly improves the visual fidelity for the Multi-Resolution Grid (MRGrid) model. "Tokyo" by Trevor Dobson (CC BY-NC-ND 2.0).

where the DRR-enhanced image is smoother without losing structural details. Most importantly, DRR demonstrates a superior ability to faithfully recover critical object details. In the highest magnification view at level 3, the traffic signs are poorly reconstructed by both NGP and MRGrid. In the NGP prediction, the signs nearly disappear into blurry artifacts, while the MRGrid struggles to recover defining features like the red diagonal line. The DRR-enhanced MRGrid, however, preserves the visual characteristics of the signs to a much greater extent, allowing them to be easily perceived and recognized.

This highlights the advantage of DRR, especially for challenging data with densely packed, high-frequency features across the spatial domain. The visual results confirm that DRR provides a significant additive benefit to embedding-based structures, improving reconstruction fidelity while preserving their critical efficiency properties. The improvement is most pronounced when a grid-based model is applied to a suitable dataset, such as a complex scene with diverse features throughout the domain for dense feature grid.

This success suggests a promising direction for future work. For data with sparser features where hash-grid models like NGP excel, it would be an interesting area of exploration to study whether applying the DRR principle to refine the hash grid could boost performance even further.

**Zero-Shot Super-Resolution Evaluation.** To further test the quality of DRR-enhanced models, we evaluate the models on the more challenging task of zero-shot ×2 super-resolution. Similar to our spatio-conditional evaluation, the INRs are trained on data that have been downsampled to half their original resolution. The models are then evaluated on their ability to reconstruct the original,

Table 11: Zero-shot ×2 super-resolution evaluation for Gigapixel images. The results for DRR-enhanced models are shown as the Δ compared to the baseline in the previous row. NGP and SIREN are shown as performance references. Green indicates an improvement.

| Gigapixel Image | Model | PSNR↑ | SSIM↑ | LPIPS↓ | Inference Time (sec) | # Params |
|---|---|---|---|---|---|---|
| Girl with a Pearl Earring | Grid | 25.80 | 0.9178 | 0.2514 | 0.88s | 4.2M |
| | + DRR | +0.13 | +0.0021 | -0.0053 | 1.16s | 4.3M |
| | MRGrid | 25.86 | 0.9188 | 0.2489 | 1.19s | 4.2M |
| | + DRR | +0.03 | +0.0003 | +0.0009 | 3.21s | 4.3M |
| Train: 11733×10000 Test: 23466×20000 | NGP | 26.31 | 0.9151 | 0.2354 | 0.90s | 5.2M |
| | SIREN | 24.81 | 0.8776 | 0.2786 | 27.82s | 2.1M |
| Tokyo | Grid | 27.15 | 0.9398 | 0.1727 | 2.12s | 4.2M |
| | + DRR | +1.11 | +0.0106 | -0.0162 | 2.56s | 4.3M |
| | MRGrid | 28.07 | 0.9492 | 0.1519 | 2.92s | 4.2M |
| | + DRR | +0.73 | +0.0057 | -0.0096 | 5.54s | 4.3M |
| Train: 10725×27308 Test: 21450×56718 | NGP | 25.60 | 0.8975 | 0.1923 | 2.23s | 5.2M |
| | SIREN | 26.47 | 0.9268 | 0.1903 | 71.28s | 2.1M |

full-resolution images, a task which requires the learned representation to be truly continuous and generalizable across scales.

The results, presented in Tab. 11, demonstrate that applying DRR consistently enhances the zero-shot super-resolution capabilities of both base models. The overall observations are consistent with the previous results with full-resolution training in Tab. 10, where DRR substantially boosts the performance of models for the complex "Tokyo" scene. For the "Girl with a Pearl Earring" image, DRR also provides a consistent, though more modest, improvement for both backbones. Notably, for the MRGrid model, DRR provides a slight improvement in this more challenging task, in contrast to the slight degradation seen in the standard reconstruction setting. These results consolidate the finding that DRR provides a consistent fidelity advantage to embedding-based models while maintaining promising inference efficiency.

As a final remark of this evaluation, the similarity of these outcomes to the standard evaluation setting shown in Tab. 10 can suggest that a ×2 downsampling factor may be too conservative to fully probe the generalization limits of INRs on such ultra-high-resolution images. Future explorations using more aggressive downsampling factors, such as ×4 or ×8, could offer deeper insights into the spatial generalization capabilities of DRR-enhanced models.

## I.2 NEURAL RADIANCE FIELD FOR NOVEL VIEW SYNTHESIS

Novel View Synthesis (NVS) is another application of INR that enables faithful and view-consistent 3D reconstruction with only 2D supervision. Neural Radiance Field (NeRF) (Mildenhall et al., 2020) is one seminal INR formulation for NVS, and we demonstrate the effect of DRR on NeRF models for two NeRF synthetic datasets.

We integrate DRR with three embedding-based NeRF models. The same models in the Gigapixel image representation tasks are adapted, including the basic 3D feature grid INR (Grid) and a multi-resolution feature grid model (MRGrid). In addition, a plane-based decomposition model (TriPlane (Chan et al., 2022)) is included as a competitive embedding structure for 3D data. For this experiment, we apply a simplified version of DRR where the refiner directly enhances the base embeddings without the preprocessing steps of super-resolution or P.E. feature upsampling. All models are trained and evaluated following the implementation of the open-sourced NerfAcc repository (Li et al., 2023). We again include two baselines in Tab. 12 as references of performance: the original MLP-based NeRF (Mildenhall et al., 2020) and the Instant-NGP (Müller et al., 2022).

The results in Tab. 12 show a consistent trend: applying DRR to models built on embedding structures consistently improves both reconstruction (PSNR) and perceptual (LPIPS (Zhang et al., 2018)) metrics. Notably, this performance gain is also accompanied by a reduction in inference time for

Table 12: Performance comparison of neural radiance field (NeRF) models with DRR-Enhanced embedding structures. The performance of NGP (Müller et al., 2022) and the original NeRF model (Mildenhall et al., 2020) is presented as a reference. Values for "+ DRR" rows show the change relative to the baseline structure above. Green indicates an improvement (higher PSNR, lower LPIPS).

| Dataset | Model | PSNR↑ | LPIPS↓ | Inference Time (sec) | # Params |
|---------|-------|-------|--------|----------------------|----------|
| Lego | Grid | 33.33 | 0.041 | 48s | 33.6M |
|  | + DRR | +0.06 | +0.002 | 51s | 33.6M |
|  | MRGrid | 32.49 | 0.053 | 55s | 19.0M |
|  | + DRR | +0.31 | -0.001 | 51s | 19.0M |
|  | TriPlane | 35.32 | 0.032 | 71s | 9.4M |
|  | + DRR | +0.07 | -0.005 | 67s | 9.7M |
|  | NGP | 35.46 | 0.026 | 54s | 12.6M |
|  | NeRF | 33.69 | 0.051 | 318s | 0.6M |
| Drums | Grid | 25.31 | 0.082 | 48s | 33.6M |
|  | + DRR | -0.54 | +0.004 | 53s | 33.6M |
|  | MRGrid | 25.37 | 0.085 | 52s | 19.0M |
|  | + DRR | +0.06 | -0.001 | 50s | 19.0M |
|  | TriPlane | 25.06 | 0.104 | 71s | 9.4M |
|  | + DRR | +0.30 | -0.018 | 68s | 9.7M |
|  | NGP | 25.69 | 0.083 | 53s | 12.6M |
|  | NeRF | 25.22 | 0.089 | 277s | 0.6M |

MRGrid, as the multiple grids are refined into a single representation with multi-scale features, thereby reducing the computation and overheads involved in the interpolation functions.

Interestingly, we observe that applying DRR can degrade the performance of a simple, single-resolution Grid on the Drums scene. Similar to the MRGrid observation from Sec. I.1, we hypothesize this occurs when a base representation is already performing at its capacity limit for a given task, such as a high-resolution feature grid with 33.55 million parameters for a single radiance field. In such cases, adding more non-linear transformations via the refiner may not meaningfully extend the already-saturated features. What remains clear, however, is the main trend: for modern, scalable embedding structures like MRGrid and TriPlane, DRR is an effective tool for boosting the performance of the resulting NeRF models with minimal speed penalty.

