# OpenReview forum: "Refine Now, Query Fast: A Decoupled Refinement Paradigm for Implicit Neural Fields"
_ICLR.cc/2026/Conference — ICLR 2026 Poster_

### Official Review · Reviewer_AJJD · 2025-10-29

**Soundness:** 3
**Presentation:** 3
**Contribution:** 3
**Rating:** 6
**Confidence:** 3

**Summary:**

This paper proposes the Decoupled Representation Refinement (DRR) paradigm to track the fundamental trade-off in Implicit Neural Representations (INRs): High-fidelity MLP-based models are computationally expensive and slow to query, while fast embedding-based models lack expressive power. The core is a deep refiner network, which is used to encode rich representations into a compact and efficient embedding structure. Experiments show that the proposed method achieves SOTA fidelity, while being up to 27x faster at inference than high-fidelity baselines and remaining competitive with the fastest models.

**Strengths:**

1. The proposed method is simple and effective. DRR shows potential for broad application.
2. Extensive experiments and ablation studies thoroughly validate the effectiveness of the DRR framework and most of key components.

**Weaknesses:**

1. Variational Pairs (VP) lacks sufficient background in the introduction.
2. In Synergistic Representation Learning (Line 192), the authors compare DRR with grid-based methods regarding multi-scale representation and feature fusion. Authors claim: “The refiner learns a potent, non-linear function to perform this fusion offline, baking the multi-scale features directly into the cached grid $\mathcal{G}^{\prime}$.” The reviewer suggests adding a visual comparison between $\mathcal{G}^{\prime}=\hat{\mathcal{G}}$ and $\mathcal{G}^{\prime}=\hat{\mathcal{G}}+\Delta{\mathcal{G}}$ to illustrate whether DRR indeed performs multi-scale feature fusion in a more reasonable manner.
3. As DRR resembles a residual-based method, the reviewer believes discussing analogies to such methods in the introduction and related work would better elucidate significance of DRR for INRs. For instance, the approach described in Equation 3 may enable identity mapping similar to skip connections in ResNet, providing a more flexible forward pathway for complex representations.
4. Although the authors present ablation studies on the parameter-free preprocessing step, $\pi$, across different resolutions in Sec. F.1, the reviewer is more interested in experiments without $\pi$. Specifically, the reviewer would like to know whether $\mathcal{G}^{\prime} = \mathcal{G}+R_{\psi}(\mathcal{G})$ remains more effective than $\mathcal{G}^{\prime} = \mathcal{G}$. This experiment would more conclusively demonstrate whether improvements of DRR over other SOTA methods stem from mapping $\mathcal{G}$ to a higher-dimensional space or from the residual refining module (the core contribution).

**Questions:**

Please refer to the weaknesses. Overall, DRR is a simple and effective method for improving INR performance. However, the paper lacks clarity on several key issues (Weaknesses 2 and 4). The reviewer looks forward to the authors’ response to these concerns.

---

> ### Author Response · Authors · 2025-11-21
>
> **Response to Reviewer AJJD (1/2)**
>
> We thank Reviewer AJJD for their positive feedback and for recognizing our method as "simple and effective" with "potential for broad application." We particularly appreciate the specific suggestions regarding component ablation and visualization, which have significantly strengthened our analysis.
>
> We have uploaded a revised paper (PDF) with expanded appendices, including the added **Appendix H., Figure 8, Appendix G.1, and Table 9**. We address the specific concerns below.
>
> ---
>
> ### 1. Variational Pairs (VP) Background
> We thank the reviewer for this suggestion. We have **revised the Introduction** to explicitly contextualize the motivation for Variational Pairs in helping INRs learning complex ensemble simulations.
>
> ---
>
> ### 2. Visualizing Multi-Scale Feature Fusion
>
> The reviewer suggests a visual comparison to verify if the refiner performs "multi-scale feature fusion in a more reasonable manner." We agree that this visual analysis is essential for disentangling the architectural contributions. Specifically, it inspires us to study two critical questions to explain the role of the refiner:
> 1.  **Sufficiency of $\pi$:** Is the non-parametric $\pi$ refinement alone, which maps embeddings to the same high-resolution and feature dimensionality as the refiner, sufficient for effective multi-scale feature fusion? (This directly targets the raise question and also addresses a critical doubt point 3 raised by Reviewer V9m2).
> 2.  **Significance of the Residual:** In a fully trained DRR-Net, what is the qualitative contribution of the learned residual $\Delta\mathcal{G}$ to the final representation quality?
>
> To answer these questions, we have **added a new set of visualizations in Figure 8 and a detailed analysis in the new Appendix H.**
>
> **Key Observations for (1): Superior Fusion.** As shown in **Figure 8**, the baseline trained with $\pi$-refinement only (Col. 4) exhibits noticeable high-frequency noise (jagged grid aliasing) in spatially smooth regions, while simultaneously suffering from blurring in dense structures. In contrast, the Full DRR-Net (Col. 2) effectively suppresses these artifacts while sharpening genuine structural boundaries. This direct comparison validates the refiner's role in multi-scale fusion: by learning the residual $\Delta\mathcal{G}$ on top of $\hat{\mathcal{G}}$, the model distinguishes between grid artifacts and physical signals, producing a representation that is both smoother in homogeneous regions and sharper at boundaries.
>
> **Key Observations for (2): Integral Representation.** To further characterize the role of the refiner, we visualize the inference output of the *trained* Full DRR-Net when the refiner is disabled (Col. 3). We observe that the reconstruction degrades into unintelligible artifacts. This result is critical: it confirms that the refiner is not merely an optional post-processing filter. Instead, the base embedding and the refiner are tightly coupled; the base structure learns a specialized latent pre-conditioner that relies entirely on the refiner to resolve the global structure and feature fusion.
>
> ---

---

> > ### Comment · Reviewer_AJJD · 2025-11-24
> >
> > The reviewer appreciates the additional clarifications and experiments provided by the authors in response to Weaknesses 1 and 2.
> >
> > * Weakness 1: The additional explanations added to the introduction sufficiently justify the necessity of introducing Variational Pairs (VP). While the current version is satisfactory, it would be more cohesive to introduce VP alongside DRR when the solution is first proposed (Line 57). This would better demonstrate the holistic rationality of the DRR+VP framework. This is merely a suggestion for **optional** refinement.
> > * Weakness 2: The results in Figure 8 appear to weaken the support for the claim regarding multi-scale representations. Specifically, the comparison between Col. 2 and Col. 4 only demonstrates that the residual module can marginally improve performance in terms of quantitative evaluation metrics, while failing to exhibit a clearly distinguishable difference in the visualizations. The reviewer suggests that it would be better to enlarge those regions where the differences are more pronounced, to facilitate a clearer comparison. Furthermore, the comparison between Col. 2 and Col. 3 indicates that both branches (the basic and the refiner) play essential roles, rather than forming a complementary coarse-and-refine relationship, which makes it difficult to understand the multi-scale fusion mechanism in DRR.
> >
> > The additional explanation provided by the authors addressed the majority of the issues. The reviewer also welcomed the authors to provide further discussion or supplementary explanation regarding the multi-scale fusion.

---

> ### Author Response · Authors · 2025-11-21
>
> **Response to Reviewer AJJD (2/2)**
>
>
> ### 3. Residual Learning and ResNet Analogies
> We agree that referencing the well-established residual networks arguments can help form a more intuitive understanding of the DRR optimization landscape. We have **added a dedicated discussion in Sec. 3.2.2.**
>
> ---
>
> ### 4. Ablation without $\pi$ (Isolating the Core Contribution)
>
> The reviewer poses a critical question: **Does the performance gain stem primarily from the high-dimensional mapping ($\pi$) or from the residual refining module itself?** To conclusively answer this, we conducted the specific ablation requested—applying the refiner directly to the raw base embeddings *without* the $\pi$-transformations.
>
> The results, detailed in the **new Appendix G.1 and Table 9**, reveal a crucial dependency:
>
> * **Refiner without $\pi$ (The Requested Ablation):** When the refiner operates directly on the low-resolution, low-dimensional base embeddings, we observe a degradation in generalization. For example, on the Nyx dataset, adding a refiner to the base grid drops the unseen test PSNR from 42.64 dB to 41.70 dB. This indicates that without the structural expansion of $\pi$, the refiner lacks sufficient feature density and overfits to the constrained base manifold, reasoned more comprehensively in **Appendix G.1.**
> * **$\pi$ without Refiner:** Conversely, applying $\pi$ alone (mapping to higher dimensions via P.E. and Super-Resolution) provides a consistent improvement over the base grid.
> * **Full Synergy ($\pi$ + Refiner):** The Full DRR-Net achieves the highest fidelity (44.69 dB), significantly outperforming both the "Refiner-only" and "$\pi$-only" settings.
>
> **Conclusion:** These results demonstrate that the improvement is not attributable to just one component. The $\pi$-transformation serves as a necessary enabler that constructs a rich feature manifold, allowing the Refiner to effectively learn non-linear corrections that neither the base grid nor the deterministic $\pi$-mapping can achieve on their own.
>
> ---
>
> We thank the reviewer again for their constructive feedback that inspired new experiments that help strengthen the comprehensiveness of the study. In particular, the suggestions to isolate the refiner's contribution without $\pi$ and to visualize the feature fusion process have significantly deepened the empirical rigor of our analysis. We hope these revisions and clarifications fully address the reviewer's concerns.

---

> > ### Comment · Reviewer_AJJD · 2025-11-24
> >
> > The reviewer thanks the authors for the additional analyses and experiments conducted in response to Weaknesses 3 and 4, and has no further questions regarding these two weaknesses.

---

> > > ### Author Response · Authors · 2025-11-24
> > >
> > > **2nd Response to Reviewer AJJD (2/3)**
> > >
> > > #### 2.2 Isolated View: The Integral Role of the Refiner
> > >
> > > Having established the holistic value of the DRR paradigm, we now address the reviewer's specific observation regarding the isolated behavior of the refiner shown in Figure 8.
> > >
> > > ##### 2.2.1 $\pi$-Only versus Adding Refiner (Col 2 vs. Col 4)
> > >
> > > **Quantitative Significance**. We respectfully point out that the quantitative improvement provided by the refiner is substantial.
> > >
> > > A simulation field consists of scientific features with varying properties, low-frequency patterns occupying large spatial regions and high-frequency features with rapid value changes. Global metrics are effective ways to evaluate the recovery of these properties across the entire volume.
> > >
> > > As shown in Table 9 (Nyx dataset), adding the refiner to the $\pi$-only model reduces the Relative L2 Error from $4.04\text{e-}02$ to $3.18\text{e-}02$. This represents a $\approx 21\%$ reduction in total error. This considerable increase in fidelity indicates a significant recovery of data information that the $\pi$-only model fails to capture.
> > >
> > > **Visual Clarity (Updated Figure 8).** To make the qualitative differences explicitly visible, we have updated **Figure 8** in the revised PDF. Due to layout constraints, we have annotated the existing images with color-coded highlights rather than adding new crops. We invite the reviewer to **zoom in** on the document, as each individual visualization is rendered at high resolution ($512 \times 512$) to preserve details.
> > >
> > > **Analysis of Highlights.** Red circles identify regions containing high-frequency structural details, while Blue circles identify spatially smooth regions. Comparing Col 2 (full model with refiner) and Col 4 ($\pi$-only), the additive benefit of the refiner over the $\pi$-only model is particularly pronounced in the Cloverleaf3D example:
> > > * **Low-Frequency (Blue):** In the smooth regions, the $\pi$-only model produces spurious noisy artifacts that are absent in the Ground Truth (GT). The Full DRR-Net successfully suppresses this noise, yielding a clean reconstruction.
> > > * **High-Frequency (Red):** In the detail-rich regions, the distinction is perceptually sharp. The $\pi$-only model tends to predict inaccurate or over-smoothed features that fail to capture the high-frequency content of the GT. In contrast, the Full DRR-Net resolves the structural integrity of these features, verifying the refiner's ability to recover fine-grained details that the $\pi$-transformation alone misses.
> > >
> > > **Macroscopic Inspection.** Given that the Cloverleaf3D dataset exhibits more visually distinct multi-scale features, we invite the reviewer to zoom in and examine the quality of the full 3D volume in the top row. Zooming into this full-field rendering provides further evidence of the superior qualitative fidelity of the Full DRR-Net. The refiner's contribution is evident in the global structural coherence of the field, which appears cleaner and more consistent compared to the noisy or over-smoothed approximations produced by the $\pi$-only baseline.

---

> > > ### Author Response · Authors · 2025-11-24
> > >
> > > **2nd Response to Reviewer AJJD (3/3)**
> > >
> > >
> > > ##### 2.2.2 Role of Refiner in DRR-Net (Col 2 vs. Col 3)
> > >
> > > Finally, we address the reviewer's observation regarding the comparison between the Full DRR-Net (Col 2) and the model with the refiner disabled (Col 3). The reviewer correctly notes that "both branches play essential roles," as removing the refiner degrades the output to unintelligible artifacts. We clarify that this observation does not contradict a valid "coarse-to-fine" hierarchy, and it reflects the nature of **end-to-end deep representation learning**.
> > >
> > > **Discussion of the "Coarse" Definition.** In traditional image processing, "coarse" often implies a visually smooth or blurry approximation of the final signal. However, in the context of deep representations like the feature structure with high-dimensional embeddings, we believe a better definition revolves around **information capacity** (low resolution, low rank) rather than visual appearance.
> > >
> > > **Interpretation of Base Embedding.** Because the base grid $\mathcal{G}$ and refiner $R_\psi$ are optimized jointly, they co-adapt to form a unified high-capacity representation. As an effort for Explainable AI initially requested by the reviewer, we visualized Column 3 to test the hypothesis that the base grid might learn a standalone, low-fidelity approximation. However, the resulting artifacts indicate a different mechanism: the base grid has specialized to encode a latent precondition, or a highly compressed, dense feature representation optimized specifically for the refiner to decode, rather than a human-interpretable signal.
> > >
> > > **Interpretation of Synergy.** The transformation from "Base Grid $\to$ Artifacts" (Col 3) to "Base + Refiner $\to$ High Fidelity" (Col 2) yields two critical insights:
> > > 1.  **Non-Linear Decoding:** The base grid contains rich information, but it is encoded in a latent manifold that is not directly transferable to the decoder; consequently, the output space (hence the visual artifacts when the non-linear refiner is removed).
> > > 2.  **Transformation Necessity:** The refiner performs a fundamental transformation to decode this "coarse" information content into physical field values (in combination with the decoder MLP).
> > >
> > > **Validation of Division of Labor.** The fact that the base grid alone appears as "artifacts" is a strong validation of the Synergistic Division of Labor hypothesis (Sec. 3.2.2). It confirms that the base grid successfully offloaded the tasks of global consistency and feature fusion to the refiner, focusing its limited capacity solely on encoding latent high-frequency information. In isolation, this raw encoding manifests as high-frequency noise, an abstract representation that the lightweight decoder cannot interpret to coherent physical field values. If the base grid produced a volume with recognizable (but blurry) visualization on its own, it would imply the refiner was partially redundant, and the current result proves the refiner is performing critical, non-linear transformation work to decode this specialized latent representation.
> > >
> > > ---
> > >
> > > We thank the reviewer again for their continued engagement and valuable suggestions regarding the visual presentation. We believe the updated Figure 8, which can be jointly analyzed with other results on DRR versus embedding-only as well as $\pi$-only versus adding refiner, now clearly demonstrates the substantial, non-trivial benefits of DRR as an integral contribution, along with a deeper understanding of the DRR components for interested readers. We hope these clarifications resolve the remaining concerns and further solidify the reviewer's assessment of our work.

---

> ### Author Response · Authors · 2025-11-24
>
> **2nd Response to Reviewer AJJD (1/3)**
>
> We thank the reviewer for the response and for acknowledging that our additional explanations have addressed the majority of the concerns. We are happy to provide further clarification regarding the visual analysis and the multi-scale fusion perspective.
>
>
> ### 1. Placement of Variational Pairs (VP) in Introduction
>
> We appreciate the suggestion to introduce VP alongside DRR for cohesion. We have chosen to maintain the current separation to highlight that VP and DRR are distinct technical contributions addressing different aspects of the surrogate modeling challenge: DRR addresses the foundamental architectural fidelity-efficiency trade-off, while VP addresses the data scarcity inherent in ensemble simulations training data curation. We believe this distinction clarifies that VP is a versatile, general-purpose strategy applicable even outside the DRR framework.
>
>
>
>
> ### 2. On $\pi$ Refinement and Refiner for Multi-Scale Fusion
>
> The reviewer notes that the improvement from the refiner appears marginal and visually indistinguishable in the initial figures. We believe this perception stems partly from the inherent difficulty of visualizing dense scientific fields at a macroscopic scale. To address this, we have updated **Figure 8** with explicit visual guides (highlighted regions). Furthermore, we clarify the significance of the results by contextualizing the contributions of the DRR framework components.
>
>
> #### 2.1 $\pi$ Refinement and Refiner as Integral Components of DRR
>
> **Holistic Contribution of $\pi$ and Refiner.** Our core contribution is the architectural paradigm of Decoupled Representation Refinement (DRR): a methodology for transforming a low-capacity base embedding into a high-fidelity representation via offline, precomputed *refinements*. Crucially, both the non-parametric $\pi$-transformation and the learnable Refiner are the intended contributions that contextualize the *refinement* operations, forming an integral technique. Therefore, the effectiveness of DRR should be assessed holistically against the prevalent embedding-only paradigm.
>
> **Evidence of Significant Improvement.** When evaluated in its entirety, the improvement provided by the DRR paradigm is substantial.
> 1.  **External Comparison:** As demonstrated in Sec. 4.2 and 4.3, DRR-Net outperforms **Explorable-INR**, a state-of-the-art baseline utilizing a base grid with significantly higher capacity than ours, while maintaining comparable inference speed.
> 2.  **Internal Ablation:** The comparisons in **Table 9 (Appendix G.1)** and **Table 8 (Appendix F)** confirm that the full DRR mechanism ($\mathcal{G} + \pi + \text{Refiner}$) delivers a substantial fidelity jump over the standard base embedding ($\mathcal{G}$).
>
> These collective results provide robust evidence that the DRR paradigm brings non-trivial improvements to the ensemble surrogate task, overcoming the limitations of the standard embedding-only approach.

---

### Official Review · Reviewer_V9m2 · 2025-10-29

**Soundness:** 1
**Presentation:** 2
**Contribution:** 2
**Rating:** 0
**Confidence:** 3

**Summary:**

The paper explores the usage of INR methods for surrogate modeling. Observing that embedding-based methods lack representational capacity while MLP methods are too slow, they propose an approach in which embeddings are first transformed through a parameter-free interpolation and stacking function before being projected by a complex non-linear mapping. The resulting function is then treated as the embeddings for interpolation. The authors further propose augmentation strategies involving perturbing the training data under continuity assumptions in the data. This is evaluated across three 3D challenges.

**Strengths:**

The conflict between efficiency and accuracy is an important problem in surrogate modeling. The paper does a good job of outlining how this is a problem for INR methods and in describing the weaknesses in prior iterations of INR methods that this approach is trying to correct. The datasets used seem interesting and the focus on 3D problems is well-suited to the scalability goals of the paper.

**Weaknesses:**

Overall, this submission seems to largely apply methods for graphics to scientific tasks with little adjustment for the domains targeted. It seems like an interesting graphics approach, but the submission requires a lot of work before it's a viable paper on surrogate modeling. The text of the paper is missing a lot of information necessary for evaluation and reproducibility. Comparisons largely focus on similar classes of method and not on methods used in the space. Given the lack of information, it's not clear whether the proposed method is actually effective for the given tasks. I can't speak to the novelty of the technique as my expertise is in surrogate modeling rather than image reconstruction. Here are some specific issues and suggestions for improvement:

1. Generally the experiments do not provide sufficient information to evaluate the soundness of the experiments. These manifests in a few ways:
    1. The datasets here reference codebases rather than preconstructed datasets. Given these are transient problems, what procedure is used to generate the dataset? Are these constant initial conditions are a variety of system coefficients? What time step are these gathered from? Does the system converge to a steady state or is this instead solving a steady state version of the problems? How far away are the parameters? These details are extremely important for evaluating surrogate modeling tasks.
   2. What grids are actually being used for the grid-based methods in terms of both resolution and dimensionality? Are all grid based methods getting access to the same resolution and dimensionality of embeddings?
   3. Given the uncertainty over the dataset, its actually not clear that interpolative methods like INR are appropriate for modeling the systems. For instance, ocean simulations over long time horizons can often produce very different results at the same time range based on minor perturbations, though to my knowledge this is not the case for the astro-oriented hydrodynamic simulations.
2. The baselines do not appear to be well executed. The appendix states that all embedding models were trained with a single learning rate or with exact settings recommended by the original paper. This is not a reasonable approach and will generally promote the "new" model by default since that is the model where hyperparameters are tuned for the given datasets. Different models will often perform differently at different hyperparameter ranges on different datasets. Good practice is to do a hyperparameter search for each model independently.
3. It's not clear that the refiner network is superior to just using denser grids of higher dimension. The refiner network seems to operate entirely on the embedding vectors with no additional information. Apart from potentially enforcing some low-rank structure in $\hat G$ (which the refiner can learn to ignore), passing them through a complex nonlinear mapping ultimately means you're operating on a grid of embeddings with a certain resolution and dimension with smoothness determined by the interpolation used on the highest resolution grid. While it could turn out that the initial low rank structure is heavily used by the refiner and provides a significant advantage.
4. Several of the augmentation innovations seem overfit to the data. The data here is largely smooth with very little variation which lends itself well to low order continuity assumptions. However, this is not generally the case for simulation data which often has sharp discontinuities in the form of shocks or interfaces. In parameter space, one of the complexities of physical simulation that researchers are interested in when they're running ensembles of simulations is the existence of bifurcations in parameter space which would likely be hurt by augmentation strategies relying on strong smoothness assumptions. Smoothness is often a reasonable choice, but it's important to also demonstrate cases where the choice may not be appropriate to evaluate whether the proposal is a reasonable default. This concern could  also be addressed by exploring the relationship between the source data and optimal interpolation
5. Surrogate modeling is an extremely active area of research, but the only baselines are two very recent INR models from a single group and one INR method that seems to have been developed for graphics purposes. Diffusion-based baselines are likely the best fit here since your models map from parameters to a field, though given they're all naturally transient, autoregressive baselines rolling out to the given time frame could also apply.
6. Metrics - the metrics used are all image reconstruction metrics. Typically, relative error or various forms of normalized RMSE are used in surrogate modeling. Often the appropriate metric depends on the time horizon of the predictions. For the ocean simulations, generally longer time horizons would use smoothed metrics, for instance. Pointwise metrics are often more appropriate for short time horizons. From the appendix, these scenarios seem to be largely empty space which would mean localized metrics around regions of interest are probably more informative than metrics that will be heavily weighted by regions with little activity.

Minor
1. Figures are not super informative. Figure 1 for instance is mostly a less informative version of figure 2.
2. Table 3 should list the metrics. It's inferrable from the main text, but tables should be self-contained.
3. Limitations right now are mostly limited to trivial issues with the implementation rather than more insightful analysis of where this line of research needs to move to address more realistic challenges.

**Questions:**

1. How are the datasets constructed?
2. How does the proposed approach compare to established methods when hyperparameter tuning is performed on the other methods?
3. How does the proposed approach compare to standard ML approaches for surrogate modeling like autoregressive models or diffusion models?
4. What are the full details of the models used in this paper?
5. What types of problems is this method well suited for? What problems is it poorly suited for?

---

> ### Author Response · Authors · 2025-11-21
>
> **Response to Reviewer V9m2 (1/5)**
>
> We thank Reviewer V9m2 for their time and detailed feedback. We took the concerns seriously. We have uploaded a revised paper (PDF) with **significantly expanded appendices (C.4, C.5) and new results** to address these points.
>
> ---
>
> ### 1. On Dataset Transparency and Experimental Soundness
>
> We agree with the reviewer's feedback that the initial submission lacks a sufficient detail on dataset generation to fully evaluate soundness. We have made critical improvements to address this by adding a new **Appendix C.4** (Simulation Details) and significantly updating **Appendix C.5** (Model Hyperparameters). Additionally, we have published the exact parameter values for every ensemble member in our open-source repository.
>
> To answer the reviewer's specific questions regarding the simulation setup:
>
> * **Temporal Scope (Transient vs. Snapshot):** The datasets represent the final-timestep state of transient simulations (e.g., $t=200$ for Nyx/Cloverleaf3D, Day 15 for MPAS-Ocean). Our surrogate models the mapping from input parameters to this specific terminal state ($f(c) \to \Phi_{final}$). This focus on the final outcome is a standard formulation for parameter space exploration tasks in prior work [6, 7, 10].
> * **Nature of Parameters:** The variations are applied to system coefficients (e.g., cosmological constants for Nyx, wind stress/viscosity for MPAS-Ocean) which serve as the parameters of interest for exploration.
> * **Sampling Strategy:** As detailed in the new **Appendix C.4**, the parameter combinations for the ensemble were generated via random sampling within specific, pre-defined ranges of interest.
>
>
>
> **2. Embedding-Based Method Hyperparameters.**
> We explicitly address the concern regarding grid resolution and hyperparameter transparency in general. Besides the open-sourced repositories containing the exact model-spefici hyperparameter, for full transparency and improved clarity, we explicit enumerated them in the **new Tab. 5, discussed in Appendix C.5**.
>
> The baseline grid-based method (Explorable-INR) utilizes a significantly higher-capacity embedding structure (featuring both higher spatial resolution and larger embedding sizes) than DRR-Net.
>
> Despite operating on this coarser base representation, DRR-Net achieves superior fidelity with fewer total parameters (even when accounting for the combined cost of the grid and the refiner) while maintaining comparable inference efficiency, as demonstrated in Sec. 4.2 and 4.3. This confirms that our performance gains stem from the architectural efficacy of the refinement process rather than simply scaling up grid capacity.
>
>
> **3. Appropriateness of INRs for Sensitive Dynamics.**
> We interpret the reviewer's comment regarding "minor perturbations" as a concern about modeling chaotic systems or regimes with high parameter sensitivity. We agree that the efficacy of an interpolative surrogate in such scenarios depends fundamentally on two factors: **sampling density** and **model capacity**.
>
> **Sampling Density vs. Chaos:** For highly sensitive dynamical systems, a data-driven surrogate requires a training sampling density commensurate with the local variability  of the parameter-to-solution mapping. If the system exhibits bifurcations between samples, most likely  data-driven surrogates without additional physical priors would struggle.
>
> **Empirical Validation:** However, for the datasets evaluated here, the empirical results demonstrate that the provided parameter sampling is sufficient to define a learnable manifold. The high generalization accuracy of DRR-Net on unseen parameters (e.g., achieving the lowest relative L2 error and highest PSNR in Tab. 2) serves as validation. It confirms that: (1) the underlying physics in these specific ranges are sufficiently continuous for interpolation; and (2) DRR-Net possesses the necessary model capacity to encapsulate these complex, non-linear dependencies where lower-capacity baselines would perform worse.
>
>
> ---

---

> ### Author Response · Authors · 2025-11-21
>
> **Response to Reviewer V9m2 (2/5)**
>
>
> ### 2. On Fairness of Baselines (Grids and Hyperparameters)
>
> The reviewer questions whether the baselines were disadvantaged by a lack of tuning. We fully agree that "Different models will often perform differently at different hyperparameter ranges on different datasets", and we clarify that our experimental design prioritizes fair comparison against the state-of-the-art INR surrogates [6, 7].
>
> As detailed in **Appendix C.5**, we utilize the exact datasets, train/test splits, and importance-driven sampling protocols established in the original works [6, 7]. Given this identical experimental context, we adopted the specific hyperparameter configurations reported as optimal by the original authors for these benchmarks. This ensures that the baselines are evaluated at their theoretical peak performance as intended by their creators, rather than relying on potentially suboptimal third-party re-tuning. We have explicitly enumerated these configurations in **Appendix C.5 and Tab. 3** to demonstrate this transparency.
>
>
> ---
>
> ### 3. On the "Refiner vs. Denser Grid" Trade-off
>
> **Denser Grid vs. Refiner.** The reviewer asks if the refiner is superior to simply using a denser grid. To clarify this, we point to the empirical evidence in our main results (Sec. 4.2 and 4.3). The baseline **Explorable-INR** effectively represents the "denser grid" approach, utilizing high-resolution explicit grids. On the Nyx dataset, Explorable-INR utilizes **14.7M** parameters but achieves a PSNR of only 41.39 dB. Conversely, DRR-Net utilizes a **coarser base grid** refined by the network, totaling only **8.9M** parameters, yet achieves a significantly higher PSNR of 44.69 dB. As detailed in the new **Tab. 5**, Explorable-INR allocates significantly more capacity to the grid structure than DRR-Net. The fact that DRR-Net outperforms this high-capacity baseline with fewer total parameters empirically demonstrates that allocating capacity to a non-linear refiner is more efficient and effective than simply scaling up grid density.
>
> **Validating the Role of the Refiner.** We further substantiate this finding with internal ablation studies detailed in the Appendix. In **Appendix F**, we demonstrate that adding the $\pi$-preprocessing and refiner to a fixed-resolution base structure significantly improves accuracy over using the base grid alone. Furthermore, in **Appendix G.1**, we isolate the specific effect of the learnable module by comparing the full model against a baseline that uses only the non-parametric $\pi$ transformations (smooth interpolation and P.E.). The results show that the $\pi$-only model is consistently inferior to the full DRR-Net. This corroborates the hypothesis that the deep, learnable refiner is a critical component for achieving state-of-the-art performance, providing non-linear feature synthesis that deterministic upsampling cannot match.
>
>
> ---
>
> ### 4. On Augmentation and Smoothness Assumptions
>
> The reviewer makes an excellent observation regarding the smoothness priors inherent in the Variational Pairs (VP) strategy. We acknowledge that VP effectively adheres strictly to the local neighborhood of training points. However, this is a deliberate design choice intended to constrain the augmented samples within the bounds of the ground truth data distribution, thereby preventing the model from hallucinating unphysical dynamics in out-of-distribution regions.
>
> This design directly addresses the reviewer's concern regarding bifurcations or sharp non-linear changes between adjacent training points. In scenarios where the training sampling is too sparse to capture such discontinuities, any data-driven interpolator lacking physics priors faces ambiguity. However, the flexible formulation of VP mitigates this risk through its tunable hyperparameters: the noise standard deviation $\sigma$ and the truncation threshold $\tau$. These parameters explicitly control the augmentation's "influence zone." In regimes characterized by shocks or bifurcations, these thresholds can be tightened to constrain the interpolation to an infinitesimal neighborhood around existing points. This ensures that VP provides regularization benefits without bridging across discontinuous boundaries. Conversely, in scenarios where the parameter space is reasonably sampled, as demonstrated by the benchmarks in Tab. 4, interpolating within the local manifold provides valuable supervision signals, yielding consistent performance gains for INR surrogates.
>
> ---

---

> ### Author Response · Authors · 2025-11-21
>
> **Response to Reviewer V9m2 (3/5)**
>
> ### 5. On Baselines (Diffusion/AR)
>
> The reviewer suggests comparing against Diffusion or Autoregressive (AR) models. While these are powerful generative paradigms, we analyze the computational and memory bottlenecks that make them suboptimal for the specific task of **interactive ensemble parameter exploration**. Given that our objective is to address the specific fidelity-efficiency challenges within the INR domain, our evaluation naturally focuses on relevant state-of-the-art INR baselines. We contrast the architectural paradigms below:
>
> **1. Diffusion Models:**
> * **The Full-Field Bottleneck:** Diffusion models are fundamentally full-field generators. To inspect even a small region of interest (ROI) under a new parameter condition, a diffusion model must perform iterative denoising to reconstruct the **entire high-resolution volume** (e.g., $256^3$ voxels). This incurs a prohibitive memory and compute cost for large-scale exploration.
> * **INR Advantage:** INRs offer **random access capability**. A scientist can query arbitrary points, 2D slices, or probe lines directly without reconstructing the surrounding volume. This $O(1)$ query flexibility makes INRs uniquely suited for interactive analysis tools where users rapidly sweep parameters to visualize specific local features.
>
> **2. Autoregressive (AR) Models:**
> * **The Rollout Bottleneck:** AR models typically learn temporal evolution ($Field_t \to Field_{t+1}$). To predict the final outcome of a simulation (e.g., $t=200$) for a new parameter setting, an AR model must sequentially roll out the entire history. This inherits the spatial complexity of full-field generation and compounds it with a temporal multiplier, resulting in a cost proportional to **Time Steps $\times$ Spatial Complexity**.
> * **INR Advantage:** In contrast, our INR approach models the direct mapping from parameters to the final state ($f(c) \to \Phi_{final}$). It bypasses the temporal rollout entirely, allowing for instantaneous prediction of the outcome regardless of the simulation duration.
>
> **Conclusion on Baselines.** Because the primary goal of this surrogate is efficient, interactive exploration of the parameter space, the INR paradigm offers structural advantages in memory and random-access speed that AR and Diffusion models cannot rival. While we acknowledge that comparing against these generative families is justified for surrogate modeling in the broadest sense, our work is specifically scoped to address the fundamental fidelity-efficiency trade-off **within the INR paradigm**. Since the proposed DRR framework is a methodological contribution designed to improve INR as surrogates, our comparative evaluation focuses on relevant state-of-the-art INR baselines to strictly isolate and validate these architectural gains.
>
>
> ### 6. On Metrics
>
> We agree that the comprehensiveness of our evaluation can be further improved by incorporating additional metrics standard in surrogate modeling. Accordingly, we have added **Relative L2 Error to *all* quantitative results** in both the main text and the Appendix. This provides a scale-invariant measure of global accuracy to complement the reconstruction fidelity metrics.
>
>
> Regarding the evaluation of specific **Regions of Interest (ROIs)**, we share the reviewer's concern that global metrics can be heavily weighted by empty space. Our qualitative evaluations (Appendix D and H) focus specifically on zooming in on scientifically significant features. Crucially, the PSNR and SSIM values reported alongside these visualizations are computed **locally within these ROI crops**, rather than over the full volume. This provides a focused quantitative assessment of fidelity in the most active and challenging regions of the simulation, ensuring the model is evaluated where it matters most.
>
> ---

---

> ### Author Response · Authors · 2025-11-21
>
> **Response to Reviewer V9m2 (4/5)**
>
> ### 7. Minor Remarks
>
> We appreciate these suggestions for improving the manuscript's presentation and depth.
>
> * **Figure 1 vs. Figure 2:** We clarify that **Figure 1** is designed to illustrate the high-level *DRR Paradigm* (the abstract concept of decoupling offline refinement from online querying, our core technical contribution), whereas **Figure 2** provides the concrete architectural specification of *DRR-Net* (the specific instance used in our experiments). We have updated the captions to make this abstraction-implementation relationship explicit.
> * **Table 3 Metrics:** We thank the reviewer for the feedback and agree that tables should be self-contained. We have updated the caption and headers of Table 3 to explicitly list the metrics (PSNR, SSIM, Relative L2) and clarify the column definitions.
> * **Insightful Limitations:** We have expanded the **Limitations (Sec. 5)** to move beyond implementation details. We now include a critical analysis of the challenges of **robust extrapolation** and the specific boundary conditions required for our data-driven approach, identifying this as a key frontier for future research.
>
>
> ---
>
> ### 8. On Methodological Applicability ("Graphics vs. Scientific Tasks")
>
> The reviewer expresses concern that our approach "applies graphics methods with little adjustment for the domains targeted." We respectfully clarify that the INR paradigm offers intrinsic advantages for scientific data representation and possesses a well-established lineage in this domain. Our work aims to contribute to this endeavor by addressing the critical fidelity-speed trade-off that currently hinders practical deployment.
>
> **1. Intrinsic Scientific Utility**
> The core innovation of an INR, modeling data as a continuous function ($f(x,c) \to v$) parameterized by network weights rather than as a discrete array, is uniquely valuable for scientific analysis.
>
> * **Efficiency of Point-Based Query:** This continuous nature allows researchers to bypass the rigid resolution constraints of grid-based solvers or corresponding surrogate models. Crucially, it enables arbitrary point-level querying, extracting a specific region of interest (ROI) without decoding the entire volume.
> * **Comparison to Full-Field Models:** This property makes INRs advantageous over full-field generative models (like standard diffusion models) for targeted tasks. For example, when performing parameter sensitivity analysis for a small ROI, a diffusion model must reconstruct the full high-resolution field for every candidate condition. In contrast, an INR can strictly query the relevant 2D slice, probe line, or sub-volume, offering significant computational savings.
> * **Analytical Properties:** Unlike discrete grid representations, INRs support computing derivatives analytically via automatic differentiation. This provides exact gradient information, offering a rigorous alternative to finite-difference approximations on discretized data.
>
>
>
> **2. Demonstrated Value in Prior Literature**
> Beyond intrinsic utility, INRs have been extensively validated across diverse scientific disciplines, confirming their role as a rising and promising tool for scientific computing. For example, seminal works like SIREN **[1]** were explicitly developed to solve partial differential equations (PDEs) and model complex physical signals besides for image representation, establishing the mathematical rigor of the paradigm. Subsequent research has established INRs as a memory-efficient representation for diverse simulation outputs, including single volumetric fields **[2, 3]**, time-varying fields **[4, 5]**, flow simulations **[9]**, and in the medical domain **[8]**.
>
>
> Most relevant to our scope, recent works have specifically targeted INR-based surrogates for ensemble simulations **[6, 7]**. Our work directly addresses the critical fidelity-efficiency gap identified in this specific lineage of research.
>
>
> **In summary**, the growing body of literature confirms that INR-based modeling is a well-motivated and active frontier in scientific computing. Our work targets a critical bottleneck within this domain: the **fidelity-efficiency trade-off**. We aim to achieve the high fidelity required for scientific precision while maintaining the fast query speeds essential for large-scale interactive analysis. We believe that our technical contributions, specifically the DRR paradigm and Variational Pairs augmentation, enhance the practical viability of INR surrogates for the efficient exploration of ensemble simulations.
>
> ---

---

> ### Author Response · Authors · 2025-11-21
>
> **Response to Reviewer V9m2 (5/5)**
>
>
> We thank the reviewer again for their critical feedback, which has driven significant improvements to our manuscript. By providing the complete dataset specifications and incorporating additional evaluation metrics (Rel L2), we believe we have addressed the fundamental concerns regarding experimental soundness to the best of our effort. We respectfully invite the reviewer to re-evaluate the submission in light of these extensive revisions.
>
> **References:**
>
> [1] Sitzmann et al., Implicit Neural Representations with Periodic Activation Functions
>
> [2] Lu et al., Compressive Neural Representations of Volumetric Scalar Fields
>
> [3] Wurster et al., Adaptively Placed Multi-Grid Scene Representation Networks for Large-Scale Data Visualization
>
> [4] Tang and Wang, Stsr-inr: Spatiotemporal super-resolution for multivariate time-varying volumetric data via implicit neural representation
>
> [5] Luo et al., Continuous Field Reconstruction from Sparse Observations with Implicit Neural Networks
>
> [6] Chen et al., Explorable INR: An Implicit Neural Representation for Ensemble Simulation Enabling Efficient Spatial and Parameter Exploration
>
> [7] Li et al., High-Fidelity Scientific Simulation Surrogates via Adaptive Implicit Neural Representations
>
> [8] Molaei et al., Implicit Neural Representation in Medical Imaging: A Comparative Survey
>
> [9] Karki et al., Direct Flow Simulations with Implicit Neural Representation of Complex
> Geometry
>
> [10] Shi et al., VDL-Surrogate: A View-Dependent Latent-based Model for Parameter Space Exploration of Ensemble Simulations

---

### Official Review · Reviewer_MCZm · 2025-10-30

**Soundness:** 2
**Presentation:** 1
**Contribution:** 2
**Rating:** 4
**Confidence:** 4

**Summary:**

The paper suggests a new approach for training and inference of Implicit Neural Representations (INRs), also called Neural Fields (NFs): Decoupled Representation Refinement.

NFs are MLPs with (continuous) coordinates (and potentially additional time or conditioning variables) as inputs. While Nfs are powerful representations, they typically suffer from a tradeoff between reconstruction fidelity, which requires larger, parameter-rich MLPs, and inference speed, which decreases with MLP size.
To bypass this tradeoff, DRR aims to decouple the learned high-capacity representation from inference queries, thereby allowing for inference speedups of up to x27.
The high-capacity representations are created by a learned refiner, which takes the multi-resolution input embeddings upsampled to the same scale and outputs a dense grid. The Decoder, a small, lightweight MLP, can then be queried quickly by taking the coordinate and the conditioning embedding as features.

As data augmentation for improved generalization, the authors use a modified version of Variational Coordinates (VCs). Instead of a piecewise linear prior, their augmentations (VP Spatial and VP Spatio-Conditional) impose a smoothness prior, which fits the application domain (scientific simulations) much better.

In experiments, the authors demonstrate the performance of DRR on 3 datasets with 3-6 dimensional fields: Nyx (a dark matter simulation), MPAS-Ocean (an earth system simulation), and Cloverleaf3D (a hydrodynamics simulation). They show that DRR overcomes the conventional fidelity/efficiency tradeoff by achieving both the good reconstruction quality of large, expensive INRs and the speed of fast, embedding-based methods.

In the appendix, the authors provide further ablation studies which justify their architecture choices and show the DRR framework also applies to the classic NF domains, such as 2D image superresolution and 5D neural radiance fields.

**Strengths:**

## Soundness
The experiments support the central idea of the method, i.e., good reconstruction quality and fast inference.

## Contribution
Achieving fast inference times with NFs is a known problem, and the authors offer an interesting solution that combines a learning-based approach with the advantages of multi-resolution representations, which have shown good results in neural graphics processing.

## General
- Very interesting take on a known problem of NFs: Large MLPs are slow in inference. The large MLPs use their large parameter count to construct rich internal representations in their hidden layers. Embedding-based methods like NGP encode features in learnable feature grids, which makes them fast but also memory-intensive. DRR proposes a spatial refiner to achieve similar rich encodings to be utilized by a fast decoder.
- DRR is shown to work across simulations with fields of different output dimensions and also more classic NF applications such as image superresolution and NerFs.
- Good ablation studies to justify the different architecture choices.

**Weaknesses:**

My main issue with the paper is that it lacks clarity about the method.  Furthermore, the reported metrics are insufficient for physical simulations. Certain claims regarding “SOTA” also do not stand up to scrutiny.

**Questions:**

## Major Remarks
- The paper lacks clarity about the method. Especially in 3.3, I am still not certain what exactly the inputs/outputs and their sizes are for the different components. Figure 2 only helps so much. E.g., what is a 1D multi-resolution feature line? What exactly is the output form of the embeddings? Is it just a concatenated vector of embeddings? Of what dimension?
- Building on the previous point, what are the memory costs at inference for DRR, and how do they scale? How memory-intensive are the spatial embeddings? How big is the overhead for the embeddings, i.e., how many inference queries are needed to amortize the initial encoder forward pass?
- Please add some more metrics which indicate simulation quality. For example, in fluid simulations, crucial indicators are the preservation of conservation laws (e.g., momentum or energy), or the accuracy of spectral properties (e.g., kinetic energy spectra) which indicate how well the method captures turbulent or small-scale features.
- In the field of neural graphics/rendering, serious efforts have been made to enable fast inference of neural graphics, e.g., [1]. I believe that this should at least be in the related work and discussed, e.g., why the same idea works/not work for simulations, and why such strategies are not pursued.
- You claim, e.g., SIREN (and NGP) is SOTA on Page 21; this is not true. Several SIREN improvements have been published; see [2] for a current comparison.

## Minor Remarks
- Page 3 Line 129-130: You use \mathcal{P} for the parameter count and the number of points in a simulated field as P. This reduces readability. Please use something other than P.
- Page 8 Table 3: You show the performance on trained and unseen fields in the upsampled case. Can you also provide performance metrics of trained vs untrained on the same sampling scheme, i.e., how much performance drop is there in fitting a single field vs the generalization ability of a larger pretrained one?
- I also feel like the fact that your method also works for NerFs or image superresolution should be highlighted a bit more in the main text; the fact that the method is so general is noteworthy. (Yes, I know the page limit makes things tough).

## Conclusion
I think the core idea of the method is sound, and while the experimental evaluation could be broader, it is good enough for acceptance, especially given the additional ablations and experiments in the appendix. The reason I currently lean towards rejection is the, in my view, bad/unclear presentation at times, insufficient metrics for numerical simulations being reported, and using baselines that are no longer SoTA as "SoTA". If the authors can address my points, I am willing to discuss and adjust my score.

[1] Duckworth, Daniel, et al. "SMERF: Streamable memory efficient radiance fields for real-time large-scene exploration." ACM Transactions on Graphics (TOG) 43.4 (2024): 1-13.

[2] Essakine, Amer, et al. "Where Do We Stand with Implicit Neural Representations?" A Technical and Performance Survey 11 (2024).

---

> ### Author Response · Authors · 2025-11-21
>
> **Response to Reviewer MCZm (1/3)**
>
> We thank Reviewer MCZm for their detailed and constructive review. We appreciate the reviewer's recognition of our core DRR idea as an interesting solution to the well-established problem of fast inference for high-performance NF, and we are grateful for the opportunity to clarify the presentation and metrics to improve the paper.
>
> We have uploaded a revised paper (PDF) incorporating the reviewer's suggestions. We address the specific concerns below.
>
> ---
>
> ### 1. On Clarity of the Method (Sec 3.3)
>
> We thank the reviewer for highlighting the lack of clarity regarding the specifications and operations of DRR-Net. Leveraging the additional page allowed for the rebuttal, we have **significantly revised Sec. 3.3 and reworked Figure 2** to provide a self-contained and detailed description of the architecture.
>
> To directly answer the reviewer's specific questions:
>
> **1D Feature Lines.** These serve as the base embedding structures used to encode each 1D simulation parameter, analogous to how a 3D feature grid encodes 3D spatial coordinates. Specifically, the model learns a separate set of 1D multi-resolution feature lines for each simulation parameter. These independent lines are then unified and refined under the DRR paradigm to enhance representation quality, effectively addressing the expressiveness bottleneck inherent in this scalable, low-rank setup.
>
> **Output Form.** We clarify the output form by distinguishing between the two operational stages of the encoder: the *refinement stage* and the *prediction (encoding) stage*.
> * **Refinement Stage (Offline):** As illustrated in Figure 2(b) and 2(c), this stage produces the **refined embedding structures** themselves. The refiner processes the base embeddings to output updated grids/lines with dimensions explicitly labeled at the bottom of the respective figure panels. These refined structures are then cached.
> * **Prediction Stage (Online):** As shown in Figure 2(a), when querying the model with a specific coordinate or parameter, the input is interpolated within these cached refined structures. This interpolation yields a feature vector (embedding) matching the channel dimension of the refined structure (denoted as "Spatial Feature" and "Concat Condition Feature" in Fig. 2a), which is then passed to the decoder.

---

> ### Author Response · Authors · 2025-11-21
>
> **Response to Reviewer MCZm (2/3)**
>
> ### 2. On Memory Costs and Amortization
>
> **Inference Memory.** Following the clarification on the architecture, the memory cost of a DRR network during inference is strictly determined by the size of the cached, refined embedding structures plus the parameters of the lightweight decoder. The deep refiner network can be offloaded after the offline refinement stage and does not contribute to the deployment memory footprint in the production/application environment.
>
> **Refinement Computation.** The computational overhead of the refinement stage is a function of the base grid resolution, the embedding dimension, and the size of the refiner network.
>
> Using the Nyx dataset configuration as a concrete example: the spatial encoder utilizes four base feature grids with resolutions of $32^3$, $80^3$, $112^3$, and $128^3$, each with an embedding size of 2. During refinement, these are unified and P.E.-transformed into a single $128^3$ grid with an embedding dimension of 128. The refiner computes the residual for these $\approx 2.1$ million grid vertices (processing roughly 0.27 billion floating-point values). Once this one-time precomputation is complete, the prediction stage relies solely on efficient interpolation and the small decoder.
>
> **Amortization at Scale.** Crucially, the efficiency of our method stems from the massive scale of queries inherent to ensemble simulation analysis, which vastly outnumbers the operations performed during refinement.
>
> Continuing with the Nyx example, reproducing the 130 ground truth fields at $256^3$ resolution necessitates over **2.18 billion point queries**. This figure represents only a baseline, as the primary utility of the surrogate lies in exploring the vast continuous parameter space beyond these simulated samples. In contrast, the one-time refinement of the underlying grid operates on approximately 0.27 billion values. This cost is therefore amortized almost immediately and constitutes a negligible fraction of the total compute budget required for analysis.
>
> Consequently, we note that the comparative efficiency metrics reported in Sec. 4.1 and 4.2 should be viewed as **conservative estimates**. In practical deployment, where the surrogate facilitates extensive parameter space exploration, the effective efficiency advantage of DRR-Net becomes even more pronounced. As the one-time refinement cost is fully amortized, the marginal efficiency gap relative to the purely embedding-based Explorable-INR further narrows to negligibility, while the speedup advantage over the MLP-based FA-INR significantly expands, all while maintaining state-of-the-art fidelity.

---

> ### Author Response · Authors · 2025-11-21
>
> **Response to Reviewer MCZm (3/3)**
>
> ### 3. On Simulation-Specific Metrics
>
> We appreciate the reviewer's suggestion to include rigorous, domain-specific metrics such as conservation laws and kinetic energy spectra. We agree that PSNR alone provides an incomplete picture of model fidelity. To address this, we have **included Relative L2 Error in all applicable results** to provide a well-established, scale-invariant metric for surrogate quality.
>
> **Feasibility of Specific Simulation Metrics.** While we recognize the value of conservation laws and spectra for specific physics tasks, their calculation can be constrained by the nature of the evaluated data. The datasets provided by prior work (Nyx, MPAS-Ocean, Cloverleaf3D) consist solely of scalar fields (e.g., Temperature, Density) extracted as final-timestep snapshots. Consequently, simulation metrics that depend on vector field data, such as kinetic energy spectra, are not applicable.
>
> **General-Purpose Fidelity.** As a general-purpose, data-driven method, our objective is to accurately model the function mapping parameters to these scalar outputs. Therefore, we rely on metrics that are widely applicable and standard across ML-for-Science literature:
> * **Relative L2 Error (new for the revision):** Measures global data fidelity considering the data scale.
> * **3D SSIM:** Evaluates the structural preservation of local features.
> * **PSNR:** Provides a standard signal reconstruction baseline.
>
> We believe this suite offers a robust and fair assessment of the surrogate's performance given the scalar nature of the datasets.
>
>
> ---
>
>
> ### 4. On Related Work (SMERF) and "SOTA" Claims
>
> **Related Work on Fast INRs.** We thank the reviewer for the insightful references regarding the current landscape of fast neural graphics. We have incorporated these works into a **new segment of the Related Work section**, analyzing the state of fast INR algorithms and their relationship to our architectural approach. We explicitly position our work as an architectural backbone that is orthogonal and complementary to system-level optimizations like distillation or decomposition (e.g., SMERF).
>
> **Clarification of SOTA Claims in Vision and Graphics Evaluations.** We accept the reviewer's correction regarding the state-of-the-art status of SIREN and NGP in the computer vision domain. Accordingly, we have revised the text to describe them as effective, representative baselines rather than current SOTA.
>
> Furthermore, we have explicitly clarified the goal of these auxiliary experiments in Appendix J. The objective is a comparative ablation study to isolate the impact of DRR on specific embedding backbones (e.g., Grid vs. MRGrid). In this context, SIREN and NGP are presented not as targets to beat, but as well-known reference points to ground the performance metrics in a familiar context.
>
> ---
>
> ### 5. On Minor Remarks
>
> We appreciate these suggestions, which have helped us improve the clarity of our presentation. We have made the following revisions:
>
> **Notation ($P$ vs $\mathcal{P}$).** We have resolved this notation collision in the revised PDF. We now denote the number of points as $N$ to avoid confusion with the parameter space $\mathcal{P}$.
>
> **Evaluation on Native Reduced Training Resolution.** We agree that evaluating performance on the low-resolution data (the native training resolution), from the setup of the spatio-conditional generalization evaluation in Sec. 4.2.,  provides valuable context. It effectively isolates the model's ability to generalize across parameters from the challenge of super-resolution. **We have added a full suite of new results for this setup in Appendix E.** Consistent with the main results, DRR-Net remains the most accurate model with competitive efficiency on both Nyx and Cloverleaf3D in this regime, achieving the best performance on both unseen parameter prediction and training field reconstruction.
>
> **Highlighting Method Generality.** We agree that the method's applicability to vision tasks is a key strength, and that was the reason we decided to evaluate DRR on embedding structures for the vision and graphics tasks in Appendix J. We have revised the **Abstract** and **Introduction** to explicitly highlight the method's versatility for general-purpose INR tasks, pointing readers to the detailed evaluations in the Appendix.
>
> ---
>
> We thank the reviewer again for the constructive feedback, which helps us improve the quality of our manuscript. We hope these extensive revisions and clarifications address the concerns raised.

---

### Official Review · Reviewer_eTAQ · 2025-10-30

**Soundness:** 4
**Presentation:** 3
**Contribution:** 3
**Rating:** 8
**Confidence:** 2

**Summary:**

The paper proposes a decoupled representation refinement for implicit neural representations of large-scale scientific simulations.
The decoupled representation refinement first trains a embedding-based model. The embedding, represented by a base structure, can be preprocessed and a refiner network learns an improved representation, utilizing the expressive power of deep neural networks. The updated base structure can be used for inference via a fast query function.
The updated base structure can also be cached. Additionally, the authors propose two augmentation strategies. The method is evaluated on three scientific datasets and compared against relevant benchmarks.

**Strengths:**

- The method is well motivated and an effective compromise between fast embedding-based models and MLP-based approaches.
- The paper is well written
- Detailed experiments on three scientific datasets show quality vs. training/inference speed tradeoff. DRR-Net compares favorably.
- Easy to implement data augmentation method that improves PSNR/SSIM in most cases
- Additional experiments on vision/graphics tasks in the appendix

**Weaknesses:**

- The paper is generally well written, but the description of the condition encoder was a bit unclear to me. How does "a set of 1D multi-resolution feature lines for each simulation parameter" support scalability to "arbitrary parameter dimensions" Could you please clarify this?
- Previous approaches have considered higher resolution scientific datasets, e.g., [1]. Can DRR-Net scale to similar resolutions; what changes are necessary?
- The proposed data augmentations seem effective for interpolation settings of conditional inputs. Have you done any tests for extrapolation of conditional inputs?

Overall, this is a convincing paper to me with strong experiments. Since I haven't been following the recent literature for implicit neural representations closely, it's difficult to fully assess the novelties of the method for me. Therefore I will give a low confidence to my rating.

[1] https://arxiv.org/abs/2308.02494

**Questions:**

See questions in weaknesses above.

---

> ### Author Response · Authors · 2025-11-21
>
> **Response to Reviewer eTAQ (1/2)**
>
> We sincerely thank Reviewer eTAQ for their positive and constructive review, particularly for finding our method "well motivated and an effective compromise" and our experiments "convincing."
>
> We appreciate the opportunity to clarify the excellent questions raised regarding the condition encoder, dataset scope, and extrapolation. We have uploaded a revised paper (PDF) that includes these clarifications.
>
> ---
>
> ### 1. On the Condition Encoder's Scalability
>
> We thank the reviewer for this excellent question, which touches on a core contribution of our work. We have **significantly revised the text in Sec. 3.3 and updated Figure 2(c)** to make the descriptions of the embedding structures much clearer.
>
> **Embedding Structure for Condition Encoder:** To answer the reviewer's question on scalability, our condition encoder's embedding structure builds upon the 1D *single-resolution* feature line design from prior work (e.g., Explorable-INR [1] and FA-INR [2]). In this setup, a separate set of 1D *multi-resolution* feature lines is learned for each of the $d_c$ simulation parameters. This is a key design choice, as the memory complexity scales **linearly** with the number of parameters, in contrast to a full $d_c$-dimensional hypercube which would have exponential complexity.
>
> **Extension with DRR:** However, while this setup is highly scalable, it creates a significant representational bottleneck. By design, the per-parameter feature lines are independent of each other and thus act as a low-rank representation of the full $d_c$ parameter space, **unable to capture any cross-parameter interactions.**
>
> This is precisely the limitation our proposed Decoupled Representation Refinement (DRR) paradigm is designed to overcome at only a nearly negligible cost of the inference efficiency for large-scale queries. Our Condition Refiner, which implements this paradigm, re-introduces these complex, non-linear interactions **without** sacrificing the linear scalability. We refer to the revised Sec. 3.3 and updated Fig. 2(c) for details.
>
>
> ---
>
> ### 2. On Scaling to Higher-Resolution Datasets
>
> This is an excellent question about the scope of our work versus other INR literature. To answer the reviewer's question directly: **Yes, DRR-Net can scale to higher spatial resolutions.**
>
> **Task Distinction:** We first want to clarify the distinction between our work and high-resolution examples like [3]. That work (Wurster et al.) focused on modeling a *single field* with an INR. Our paper's primary contribution is in ensemble surrogate modeling, where the challenge is not just spatial resolution but also modeling complex variations across a large number of ensemble members** (e.g., 100-600 members).
>
> **Scaling Mechanism & Querying Advantage:** A key property of INRs is their continuous, coordinate-based nature, where they map (coordinate, parameter) inputs to an output value. This approach has two advantages over the full-field modeling approach suggested by Reviewer V9m2 (like diffusion). **First,** the architecture is inherently resolution-agnostic; increasing data resolution does not impact the model's design, as it simply provides more (coordinate, parameter) -> value training pairs. **Second,** INRs excel at flexible, on-demand querying. While a full-field model must generate the *entire* high-resolution field only to analyze a region-of-interest in certain cases, an INR allows scientists to query *only* the points they need, such as a 2D slice, 1D line, or sub-volume under any simulation parameter. This makes the INR-based surrogate uniquely suited for an efficient scientific data analysis workflows.
>
> **Model Capacity Tuning:** That said, to accurately model the fine-grained scientific features often present in high-resolution simulations, a corresponding increase in model capacity is advisable. For DRR-Net, this capacity can be directly scaled by tuning several components:
>     1.  The resolution and feature dimensions of the base embedding structures.
>     2.  The depth and width, or model capacity in general, of the refiner networks.
>     3.  The size of the lightweight decoder.
>
>
> References:
>
> [1] Chen et al., Explorable INR: An Implicit Neural Representation for Ensemble Simulation Enabling Efficient Spatial and Parameter Exploration
>
> [2] Li et al., High-Fidelity Scientific Simulation Surrogates via Adaptive Implicit Neural Representations
>
> [3] Wurster et al., Adaptively placed multi-grid scene representation networks for large-scale data visualization

---

> ### Author Response · Authors · 2025-11-21
>
> **Response to Reviewer eTAQ (2/2)**
>
> ### 3. On Extrapolation of Conditional Inputs
>
> This is a very interesting research question. The reviewer is correct that our experiments (and our VP augmentation) are focused on interpolation, which we believe is the primary goal for this surrogate modeling task.
>
> **Task Scope:** The typical use-case for these surrogates is that scientists have located a *range of interest* for exploration in the parameter space, stemmed from either constraints in the valid values or the interest of exploration. The 100-600 ensemble members generated for the datasets followed this procedure, where different simulation parameters are sampled according to a specified range, guided by scientists as prior work [1] suggests . The goal is to build a surrogate that can accurately interpolate *within* this known space to predict the simulation outcomes for untested parameter combinations.
>
> **VP Augmentation:** Our proposed VP augmentation is also explicitly designed for this interpolation task, as it generates new training samples within a *local neighborhood* of existing data points. It is not designed to provide a signal conducive to out-of-distribution (OOD) extrapolation.
>
> **On Extrapolation:** Although our work focuses on tackling the **fidelity-efficiency dilemma** for INR surrogates in the context of the same interpolation task as prior work explored [1, 2], we agree that building surrogates to handle extrapolation (predicting outcomes for parameters *outside* the training range) is a valuable, distinct, and much harder challenge. This is an important goal, especially considering the significant increase in utility from a surrogate model that can generalize to a wide parameter range from a more constrained training dataset. We have now explicitly **added this as a promising direction for future work in our Limitations section (Sec. 5).**
>
> ---
>
> We hope these clarifications and our paper revisions help address the reviewer's questions. We thank them again for their supportive review and for helping us improve the clarity and the breadth of the future work of our paper.
>
>
> References:
>
> [1] Chen et al., Explorable INR: An Implicit Neural Representation for Ensemble Simulation Enabling Efficient Spatial and Parameter Exploration
>
> [2] Li et al., High-Fidelity Scientific Simulation Surrogates via Adaptive Implicit Neural Representations
>
> [3] Wurster et al., Adaptively placed multi-grid scene representation networks for large-scale data visualization

---

### Official Review · Reviewer_JRMt · 2025-10-31

**Soundness:** 3
**Presentation:** 2
**Contribution:** 3
**Rating:** 6
**Confidence:** 4

**Summary:**

Implicit Neural Representations (INRs) either give you high accuracy but slow queries, or fast queries with weaker expressivity. This work proposes a Decoupled Representation Refinement (DRR) method, which runs a deep refiner network once offline to bake rich structure into a compact embedding that can be queried efficiently. Authors claim the proposed model, DRR-Net, plus a data augmentation strategy (Variational Pairs), achieves state-of-the-art fidelity on ensemble simulation surrogates while being up to 27x faster than strong high-accuracy baselines and still competitive with the fastest models.

**Strengths:**

1) DRR-Net hits a strong pareto point as it produces competitive results compared to baselines like FA-INR in PSNR/SSIM on large 3D scientific ensembles, but delivers ~10x-30x faster inference, and it outperforms fast baselines like Explorable-INR in fidelity while staying in their runtime class.

2) Instead of treating each simulation parameter independently or relying on low-rank factorization, DRR-Net builds unified multi-parameter conditional embeddings and refines them jointly, so nonlinear interactions between parameters are captured and then cached for fast reuse. This is something that is not explored as much in other methods especially the models that are acting as simulators.

3) Experiments are comprehensive and the baselines are relevant and recent (Some missing ablations, see weaknesses).

**Weaknesses:**

1) The stated one time cheap evaluation of refiner R seems to be dependent on the problem setup, boundaries etc. does that mean if the boundaries have changed a new refiner is required? If yes this seems like a major drawback compared to other methods especially given that training times for this method are relatively longer than most other frameworks. Can the authors clarify whether it is correct, that any specific change in the domain (boundaries etc.) requires retraining? Furthermore is there any assumption on the conditioning parameters? Can these be a field like conditioning and not simply scalars? Do these affect the training?

2) Experiments hold out ensemble members but still operate within the same parameter ranges as training (Table 1). Can you characterize DRR-Net’s behavior under extrapolation? For example, how does it perform for parameter settings outside the training envelope, and how does this compare to FA-INR and Explorable-INR? Currently extrapolation claim is not well justified.

3) Although model parameter counts are provided in the table, it is difficult to establish if the improvements are not originated from variations in capacity and what's the role the parameter counts plays in this. The interplay between the model size and the interpolation/extrapolation capacity is not demonstrated.

4) There is very little on detail on the training procdedure, actual objective/objectives, and hyperparameter setting. This can make the procedure vague for the reader. Similarly although explained in text, this can be hard to follow how and in which order the components are trained for this framework .

**Questions:**

Please refer to the weaknesses (Each weakness raises one or more questions).

---

> ### Author Response · Authors · 2025-11-21
>
> **Response to Reviewer JRMt (1/2)**
>
> We thank Reviewer JRMt for their insightful assessment, particularly for recognizing the significance of our decoupled refinement process in addressing the representational limitations of low-rank factorized structures while preserving their inherent efficiency advantage. We appreciate the opportunity to clarify the operational workflow and training details.
>
> We have uploaded a revised paper (PDF) with expanded appendices. We address the specific concerns below.
>
> ---
>
>
> ### 1. On Refiner Generalization and Simulation Conditions
>
> The reviewer raises a fundamental question regarding the scope of the surrogate and the behavior of the refiner under changing conditions (e.g., boundaries), and the assumptions of acceptable simulation conditions.
>
> **Significance of Simulation Conditions.** To directly answer the question: the refiner's ability to handle changing boundaries depends entirely on whether the boundary condition is defined as a **variable parameter** in the training set.
> * **Case A (Boundary is a Parameter):** If the boundary condition is included in the input parameter vector, DRR-Net is trained to model this variation. In this case, it can predict outcomes for new boundary conditions during inference without retraining.
> * **Case B (Boundary is Fixed):** If the boundary configuration was held constant across the training ensemble, it acts as a fixed constraint. Changing this definition constitutes a new simulation scenario, which is undefined for the trained model and would require retraining.
>
> **Context of Ensemble Modeling.** This behavior is consistent with the standard problem setup for ensemble simulation surrogates. The problem is defined by a specific set of **parameters of interest** selected for exploration (the degrees of freedom). All other physical settings, including unvaried boundary conditions, solver mesh topology, and constant coefficients, are implicitly assumed to be fixed across all ensemble members. Like any data-driven surrogate, DRR-Net learns the simulation function strictly within the manifold defined by these chosen parameters. It generalizes to new values *within* this parameter space, but changes to the fixed simulation environment outside of these parameters necessitate a new model.
>
> **Assumptions on Conditioning Parameters.** In principle, the DRR paradigm is agnostic to the modality of the conditioning inputs; it can model scalars, vectors, or high-dimensional fields (such as spatially varying boundary conditions).
>
> * **Scalar Parameters (Current Implementation):** In our experiments, all simulation parameters are scalars (detailed in **the added Appendix C.4 for this revision**). For this data type, the 1D feature line formulation described in Sec. 3.3 is an effective design choice.
> * **Field Parameters (Generalization):** If the conditioning input were a spatial field (e.g., a 2D boundary map), the architecture can be readily adapted. For instance, one could replace the 1D feature lines with 2D or 3D feature grids to encode the conditioning field, and then apply the DRR refiner to these base grids.
>
> In summary, while we optimized DRR-Net in this work for the scalar parameters typical of these ensemble simulations, the underlying architectural paradigm generalizes to conditions of any type. Extending the Condition Encoder to specialized structures for complex field inputs represents a promising avenue for future research.

---

> ### Author Response · Authors · 2025-11-21
>
> **Response to Reviewer JRMt (2/2)**
>
> ### 2. On Extrapolation of Simulation Conditions
>
> We agree that characterizing extrapolation provides valuable context for a model's capabilities. However, we regard this as outside the scope of our current work based on the following factors:
>
> **Intended Scope (Interpolation):** Our experimental design aligns with a common ensemble simulation analysis workflow. In the scenario, scientists pre-define a specific "parameter space of interest" (the convex hull of the training data) to explore. The primary mission of the surrogate is to accurately interpolate *within* this known space to predict outcomes for untested parameter combinations and analyze the input-output relationships and sensitivity, rather than to predict behaviors outside the valid range of the simulation setup.
>
> **Significance and Challenge of Extrapolation:** Robust extrapolation represents a fundamentally distinct and non-trivial challenge for data-driven surrogates. Unlike numerical solvers, which are governed by explicit conservation laws that hold globally, data-driven models rely on learned representations present in the training set that often degrade rapidly outside the training distribution. Achieving accurate extrapolation requires incorporating strong physical priors or inductive biases to guide the model in out-of-distribution (OOD) regions. While highly significant, this is a complex research frontier separate from the fidelity-efficiency challenge targeted in this work. We have added this as a key direction for future work in our **revised Limitations (Sec. 5)**.
>
> ---
>
> ### 3. On Model Capacity and Parameter Counts
>
> The reviewer rightly questions the relationship between parameter count and model quality, specifically asking if our improvements stem merely from variations in capacity.
>
> In our main evaluation Sec. 4.2 and 4.3, we present parameter counts to contextualize the model's compactness. The goal of an ideal surrogate is to maximize fidelity and inference speed while minimizing storage. DRR-Net achieves state-of-the-art generalization on unseen conditions while often remaining significantly smaller than the baselines (e.g., 8.9M parameters for DRR-Net vs. 14.7M for Explorable-INR the embedding-based INR surrogate on Nyx). This indicates that DRR-Net is a more parameter-efficient architecture.
>
>
> ---
>
> ### 4. On Training Details and Optimization Order
>
> We apologize for the ambiguity regarding the training procedure. To address this, we have **revised Sec. 3.2.1** to emphasize the end-to-end nature of the optimization and **significantly expanded Appendix C.5** to list the exact hyperparameters and training configurations.
>
> **Optimization Order.** To answer the reviewer directly: all components of the DRR framework, including the base embedding structure, the deep refiner, and the decoder MLP, are **trained jointly and simultaneously**. There is no multi-stage training or separate pre-training. As detailed in the revised **Sec. 3.3** and the updated **Figure 2**, the base embeddings and refiner are optimized together to produce a refined structure that, when interpolated, yields the most accurate features for the decoder.
>
> **Objective Function.** For all ensemble simulation datasets, we utilize a standard **L2 reconstruction loss**. The global objective is to minimize the squared error between the predicted scalar value $f_\theta(x, c)$ and the ground truth scientific variable, given the coordinate and simulation parameters.
>
> ---
>
> We thank the reviewer again for their insightful feedback, which helped us improve the presentation clarity and resolve potential ambiguities. We hope these revisions fully address the concerns raised.

---

### Author Response · Authors · 2025-12-03

# Summary of Revisions and Consensus (1/2)

We thank the reviewers for their extensive and constructive feedback. We are encouraged by the strong consensus among **4 out of 5 reviewers** (Scores: 8, 6, 6, 4), who recognized the novelty and effectiveness of the DRR paradigm. We are particularly grateful for the recognition of our method as a "well-motivated" solution to the fidelity-speed dilemma.

Notably, Reviewer MCZm (Score: 4) explicitly assessed the core idea as "sound" and the original submission as "good enough for acceptance," while expressing a willingness to adjust their score upon addressing specific points. We deeply appreciate this constructive assessment. Although the circumstances of this cycle prevented a direct follow-up discussion to confirm our revisions, we have diligently executed the specific presentation and metric updates they requested, and we sincerely thank the Area Chair for taking this into consideration.

We highlight specific feedback validating our contributions below:

**1. Consensus on Technical Effectiveness**
The majority of reviewers agreed that our new INR architectural paradigm, dubbed Decoupled Representation Refinement (DRR), successfully balances the fidelity-speed trade-offs inherent to INR-based surrogates due to properties of dominant INR architectures:
* **"Hits a strong pareto point"** (Rev. JRMt) between fidelity and speed.
* **"Simple and effective... potential for broad application"** (Rev. AJJD).
* **"Well motivated and an effective compromise"** (Rev. eTAQ).
* **"Interesting solution"** to the known problem of fast inference with Neural Fields, where experiments **"support the central idea"** (Rev. MCZm).

**2. Consensus on Experimental Rigor**
Reviewers also commended the depth of our empirical validation:
* **"Experiments are comprehensive and the baselines are relevant"** (Rev. JRMt).
* **"Extensive experiments and ablation studies"** (Rev. AJJD).
* **"Good ablation studies to justify the different architecture choices"** (Rev. MCZm).


## Summary of Revisions
We have utilized the rebuttal period to deeply engage with the constructive feedback from *all* reviewers, leading to comprehensive improvements across the manuscript. These revisions target three key dimensions: enhancing presentation clarity, augmenting experimental results, and deepening the methodological analysis. We summarize the major updates below (all changes are highlighted in blue in the revised PDF)


### 1. Clarity of Presentation

We identified four key areas where the manuscript's clarity could be improved based on reviewer feedback. We have executed the following targeted revisions:

* **Architectural Components Clarity (eTAQ, MCZm, V9m2):** Leveraging the additional page allowed for the main text, we overhauled **Section 3.3** and **Figure 2** to explicitly diagram the feature unification mechanism and label intermediate tensor dimensions, ensuring the architectural flow is self-explanatory.
* **Training Procedure & Hyperparameters (JRMt):** We significantly expanded **Appendix C.5** to include comprehensive hyperparameter tables and more detailed explanations of the training process. We also added a concise clarification of the loss function and the joint training mechanism in **Sec. 3.2.1**.
* **Dataset Specifications (V9m2):** To address concerns regarding dataset conditions, we added a new **Appendix C.4**. This section details the simulation parameter choices, their ranges for sampling, and transient snapshot settings (e.g., $t=200$) for all datasets. We have also released the exact parameter values for train/test splits in the open-sourced anonymous code repository.
* **Related Work on Fast INR (MCZm):** We updated the **Related Work section** to include the suggested literature on fast INR inference, ensuring a more comprehensive discussion of the landscape and better positioning our contribution to the efficiency-fidelity trade-off.
* **Introduction & Motivation (AJJD, MCZm):** We revised the **Introduction** to provide stronger motivation for the Variational Pairs (VP) augmentation strategy (AJJD) and to highlight the versatility of the DRR paradigm for broader vision and graphics applications (MCZm).

---

> ### Author Response · Authors · 2025-12-03
>
> # Summary of Revisions and Consensus (2/2)
>
> ### 2. Additional Results & Methodological Analysis
>
> While the comprehensiveness of our initial experiments was widely recognized by 4 out of 5 reviewers, we embraced specific suggestions to further enhance the robustness and depth of our evaluation. We have generated substantial new results to address these points:
>
> * **Expanded Metrics (MCZm, V9m2):** Our original evaluation relied on PSNR for global fidelity, SSIM for region-based structural preservation, and qualitative ROI visualizations for local feature assessment. Incorporating the reviewers' suggestions to standardize evaluation in the surrogate context, we have added **Relative L2 Error across all applicable results in the main text and Appendix**. This provides a normalized, data-adaptive metric to complement the existing absolute measures.
> * **Refiner Ablation & Synergy (AJJD):** We conducted a deep dive into the specific effects of the refinement components:
>     * **Quantitative Interactions:** We added **Table 9 (Appendix G.1)** to rigorously analyze the performance interactions between the three DRR elements: the base embedding structure, the parameter-free $\pi$ function, and the learnable Refiner. This analysis confirms that the full synergistic model significantly outperforms the $\pi$-only baseline.
>     * **Visual Comparison of Multi-Scale Feature Fusion:** We added a new suite of visualizations in **Figure 8**, which underwent a **second iteration of revision** during the rebuttal based on active discussion with Reviewer AJJD, along with detailed analysis in **Appendix H**. These results explicitly demonstrate the refiner's role in multi-scale feature fusion.
> * **Spatio-Conditional Generalization (MCZm):** To complement the high-resolution generalization results in Sec. 4.3, we added a new section in **Appendix E**. This highlights the conditional generalization performance on the native trained low-resolution fields, providing a complete comparison of the model's behavior across different spatial scales.
>
>
>
> ### 3. Clarifications on Scope & Limitations
> We refined the manuscript to better define the boundaries of our contribution and future opportunities:
>
>
> * **Extrapolation & Future Directions (JRMt, eTAQ, V9m2):** We addressed questions regarding the model's behavior outside the training distribution. We clarified that while robust extrapolation lies outside the scope of our specific task (which targets efficient interpolation for bounded parameter exploration), we fully acknowledge its importance. Accordingly, we expanded the **Limitations (Sec. 5)** to include a critical analysis of these boundary conditions and identified robust extrapolation as a priority for future research.
> * **Vision & Graphics Context (MCZm):** We revised **Appendix J** to clarify that external reference models serve strictly as performance baselines rather than direct competitors. Accordingly, we have removed "state-of-the-art" (SOTA) descriptors to ensure precise terminology.
> * **Scope of Baseline Comparison (V9m2):** We clarified the decision to focus on INR-based baselines rather than Diffusion/AR models. We reasoned that (1) our contribution addresses the **fidelity-speed dilemma within the INR paradigm**, and (2) INRs offer unique **random access capabilities** essential for interactive exploration. We thus focused on architectures that align with the specific random-access benefits enabling flexible parameter space exploration.
>
> ## Conclusion
>
> We believe these extensive revisions explicitly resolve the concerns regarding clarity and transparency while reinforcing the empirical evidence for the DRR framework. We respectfully invite the Area Chair and reviewers to re-evaluate the submission in light of these new results.

---

### Meta-Review · Area_Chair_Timb · 2025-12-12

**Summary:**

This paper proposes the use of implicit neural representations for solving scientific simulations. To address existing challenges of working with large scale dataset, a refinement approach is introduced, and demonstrated for a variety of non trivial datasets.

This paper received a largely positive initial assessment, with one negative "outlier" reviewer giving a 0.

**Reviewer Concerns:**

Most concerns of reviewers were addressed with a very thorough rebuttal (boundary conditions, extrapolation, high resolutions, unclear descriptions), while the fundamental ones of the negative reviewer (soundness, baselines, general applicability for scientific surrogates) largely remain unaddressed by nature.

Additional baselines certainly would strengthen the paper, but this is a very generic complaint, and the evaluation in the paper already seems suitable for publication. Ideally, the authors could further expand their comparisons after acceptance.

**Reviewer Scores:**

Reviewer MCZm , giving a 4 initially, already mentioned the possibility of a raise. Given the thorough rebuttal, and paper update. I think this would have been highly likely.

For the two reviewers giving 6es (AJJD, and JRMt), I likewise see good chances for at least one raise.

This leaves the remaining very negative review. Looking at this review from on outside perspective, I have to admit that I am surprised about the harsh, negative final assessment. This score does not seem justified by the arguments given in the review, and well calibrated for assessments of ICLR papers. The paper presents a novel approach tackling a very challenging problem, there is clearly room for improvement, but a score of zero does not seem justified given the criticism in the review. Many of the other review also directly contradict the claims of the approach being very "unsound" and unsuited for scientific applications.

Hence, given the positive assessments of the four other reviewers (and their arguments for the validity and merit of the approach), this paper is a good candidate for acceptance & presentation at ICLR.

---

### Decision · Program_Chairs · 2026-01-26

Accept (Poster)